# Random Search Neural Networks for Efficient and Expressive Graph Learning

**Michael Ito**
University of Michigan
mbito@umich.edu

**Danai Koutra**
University of Michigan
dkoutra@umich.edu

**Jenna Wiens**
University of Michigan
wiensj@umich.edu

## Abstract

Random walk neural networks (RWNNs) have emerged as a promising approach for graph representation learning, leveraging recent advances in sequence models to process random walks. However, under realistic sampling constraints, RWNNs often fail to capture global structure even in small graphs due to incomplete node and edge coverage, limiting their expressivity. To address this, we propose *random search neural networks* (RSNNs), which operate on random searches, each of which guarantees full node coverage. Theoretically, we demonstrate that in sparse graphs, only $O(\log |V|)$ searches are needed to achieve full edge coverage, substantially reducing sampling complexity compared to the $O(|V|)$ walks required by RWNNs (assuming walk lengths scale with graph size). Furthermore, when paired with universal sequence models, RSNNs are universal approximators. We lastly show RSNNs are probabilistically invariant to graph isomorphisms, ensuring their expectation is an isomorphism-invariant graph function. Empirically, RSNNs consistently outperform RWNNs on molecular and protein benchmarks, achieving comparable or superior performance with up to $16\times$ fewer sampled sequences. Our work bridges theoretical and practical advances in random walk based approaches, offering an efficient and expressive framework for learning on sparse graphs.

## 1 Introduction

Early work on random walk–based graph representations focused on using skip-gram objectives to learn node embeddings from sampled walks [1, 2]. Building on these ideas and leveraging recent advances in sequence modeling, *random walk neural networks* (RWNNs) have emerged as a powerful paradigm for modern graph learning [3–8], overcoming the limitations of message-passing neural networks (MPNNs) [9–11] and graph transformers [12–14] by representing graphs as collections of random walks processed by sequence models. This advancement aligns with the broader research goal of identifying effective and expressive methods for graph representation learning [15–17]. However, despite their success, RWNNs encounter critical expressivity challenges under realistic conditions due to incomplete node and edge coverage, limiting their capacity to capture structure even in small graphs (Figure 1). In our analysis, we establish that, under partial coverage, RWNNs are strictly less expressive than traditional MPNNs, highlighting the importance of complete coverage and bridging the theoretical expressivity of the two paradigms.

To illustrate the limitations of RWNNs, consider the graph composed of connected six-cycles and side chains shown in Figure 1. Capturing the full structure of this graph requires traversing every node and edge. However, since the node and edge cover times for a random walk can scale as $O(|V||E|)$ [18], RWNNs require either prohibitively long walks or an impractically large number of samples to guarantee complete coverage. Under realistic sampling constraints where the walk's number of steps is significantly less than $O(|V||E|)$, random walks obtain only partial graph reconstruction: as shown in Figure 1(a), subgraphs induced by short random walks can miss critical structural components,

39th Conference on Neural Information Processing Systems (NeurIPS 2025).

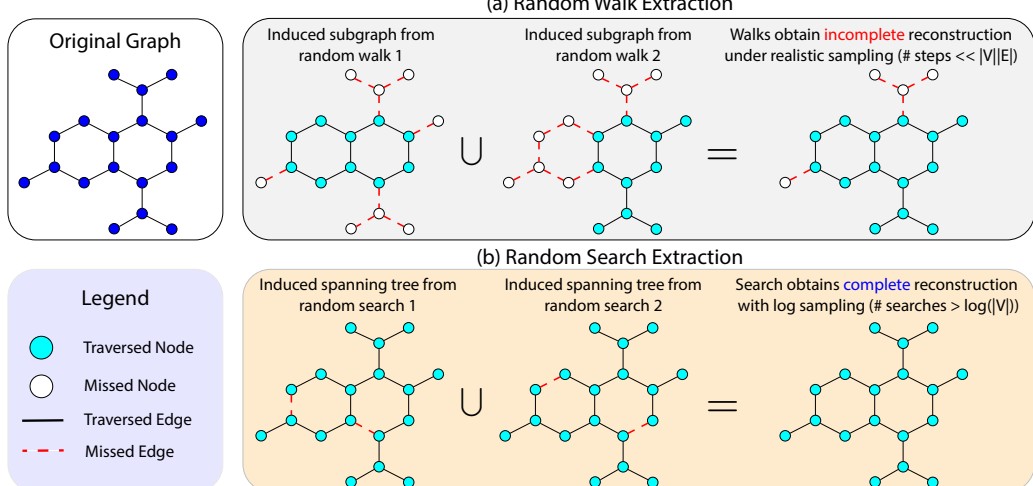

Figure 1: RWNN and RSNN coverage differences. Random walks miss critical structure under realistic sampling constraints, wheras each individual search only misses single edges in cycles, enabling complete reconstruction across logarithmic sampling in $|V|$ on sparse graphs.

such as the side chains connected to the six-cycles. This incomplete coverage significantly hinders RWNN expressivity. Current methods attempt to address this limitation through non-backtracking walks [5, 6] and minimum-degree local rules (MDLR) [7], reducing node and edge cover time to $O(|V|^2)$. Nonetheless, these approaches retain quadratic complexity with respect to graph size, making comprehensive coverage costly and impractical for even small and medium graphs.

To overcome these challenges in small and medium sized graphs, we introduce *random search neural networks (RSNNs)*, which represent graphs as collections of random searches. Critical to our analysis is the insight that subgraphs induced by searches are spanning trees as opposed to arbitrary subgraphs induced by random walks. Each spanning tree inherently ensures full node coverage, reducing the task to achieving edge coverage across the union of induced trees. Leveraging this insight, our analysis demonstrates that RSNNs require only a logarithmic number of searches for complete edge coverage, specifically in sparse graphs where such searches are computationally feasible. This is a substantial improvement over the linear number of walks required by RWNNs, assuming walk lengths scale with graph size. As shown in Figure 1(b), the union of just a few spanning trees enables complete reconstruction of the graph, including nodes and edges missed by walk-induced subgraphs. When equipped with maximally expressive sequence models, RSNNs achieve universal approximation efficiently. Furthermore, we show that RSNNs are probabilistically invariant to graph isomorphisms, ensuring their expectation is an isomorphism-invariant predictor. Empirically, we focus on sparse molecular and protein graph classification datasets, domains in which RWNNs have shown significant improvement over existing GNNs. Across both domains, we demonstrate that RSNNs consistently outperform existing RWNN approaches. In summary, we make the following contributions:

- **Characterizing RWNN Expressive Limitations.** Our analysis characterizes the expressive power of RWNNs, bridging the expressivity of RWNNs and MPNNs. We demonstrate that RWNNs under partial node and edge coverage are strictly less expressive than MPNNs, motivating the design of sampling strategies that guarantee full coverage.

- **New Model: Random Search Neural Networks.** We propose random search neural networks (RSNNs), a new approach that operates on random searches, whose induced subgraphs are spanning trees, substantially reducing the sample size required for complete node and edge coverage in sparse graphs.

- **Efficient Coverage, Universal Approximation, & Isomorphism Invariance.** We demonstrate that RSNNs can achieve universal approximation efficiently with logarithmic sampling in sparse graphs. RSNNs are also probabilistically invariant to graph isomorphims, ensuring their expectation is an isomorphism-invariant function on graphs.

- **Extensive Empirical Analysis.** Focusing on sparse molecular and protein graph benchmarks, we demonstrate that RSNNs consistently outperform existing RWNNs.

## 2 Background and Preliminaries

We establish notation for graphs and random walks and next review MPNNs and RWNNs, the primary class of models under investigation. Importantly, we later bridge the expressivity of MPNNs and RWNNs. We lastly review random walk cover times, highlighting how RWNNs require prohibitively long walks or impractically large numbers of walks to guarantee full graph coverage.

### 2.1 Notation and Random Walks on Graphs

We define a graph $G = (V, \mathbf{A}, \mathbf{X})$, where $V$ is the set of nodes, $\mathbf{A} \in \{0,1\}^{|V| \times |V|}$ is the adjacency matrix representing the set of edges $E$, and $\mathbf{X} \in \mathbb{R}^{|V| \times d}$ is the node feature matrix. For each node $i \in V$, we denote its feature vector as $\mathbf{x}_i$ and its set of immediate (one-hop) neighbors as $\mathcal{N}(i)$. We define the augmented neighborhood $\hat{\mathcal{N}}(i)$, obtained by adding a self-loop to node $i$.

A random walk of length $\ell$ on $G$ produces a sequence of nodes $W = (w_0, \ldots, w_\ell)$ by first sampling an initial node $w_0 \in V$ according to a uniform distribution $P_0$, and then iteratively transitioning to subsequent nodes by sampling neighbors according to a given random walk algorithm. We let $\mathcal{W}_\ell(G)$ denote the set of all possible random walks of length $\ell$ on $G$, and let $P(\mathcal{W}(G), P_0)$ represent a probability distribution over these walks. Lastly, we define $P_m(\mathcal{W}(G)) = \{W_1, \ldots, W_m\}$ as a realization of a set of $m$ independently sampled random walks from $P(\mathcal{W}(G), P_0)$.

### 2.2 Message-passing Neural Networks and GNN Expressivity

Standard GNNs adopt a message-passing approach, where each layer iteratively updates a node's representation by aggregating the features of its neighbors [19]. Formally, the initial message-passing layer can be defined as the following propagation rule at the node level for all $i \in V$,

$$f_{\mathrm{MPNN}}(G)_i = f_{\mathrm{agg}}(\{\mathbf{x}_j \mid j \in \hat{\mathcal{N}}(i)\}),$$

where $f_{\mathrm{agg}}$ is a permutation-invariant function. Because of this aggregation step, MPNNs incur fundamental expressivity limitations and cannot distinguish certain classes of non-isomorphic graphs [15, 20]. We compare the expressivity of GNNs by the pairs of graphs they can distinguish [21], introducing the following notation. For two GNNs $f_1$ and $f_2$, we write

$$f_2 \preceq f_1 \iff \forall G, H : f_1(G) = f_1(H) \implies f_2(G) = f_2(H).$$

Thus, any pair indistinguishable by $f_1$ is also indistinguishable by $f_2$, so $f_1$ is at least as expressive as $f_2$. The relation is strict, $f_2 \prec f_1$, if $f_2 \preceq f_1$ and there exist graphs $G, H$ with $f_1(G) \neq f_1(H)$ while $f_2(G) = f_2(H)$. $f_1$ and $f_2$ are equally expressive, written $f_1 \simeq f_2$, if $f_2 \preceq f_1$ and $f_1 \preceq f_2$. These relations coincide with notions of approximation power. For example, if $f_2 \prec f_1$, every target approximable by $f_2$ is approximable by $f_1$, and there exist targets approximable by $f_1$ but not $f_2$.

### 2.3 Random Walk Neural Networks

RWNNs are a novel class of neural network on graphs that leverage sequence models to process random walks sampled from the graph. Typically, an RWNN is characterized by four key components: (1) a random walk algorithm that generates node sequences, (2) a recording function that encodes the walks into structured representations, (3) a reader neural network that processes these representations, and (4) an aggregator that combines the representations or predictions from multiple walks.

For our analysis, we assume the following representative general version of RWNN [3–8]. Specifically, we consider the random walk algorithm as uniform random walks of fixed length $\ell$, denoted by $P_m(\mathcal{W}(G)) := P(\mathcal{W}_\ell(G), \mathbb{U}(V))$, where $\mathbb{U}(V)$ denotes the uniform distribution over $V$. Given a sampled walk $W \in P_m(\mathcal{W}(G))$, we define the recording function $f_{\mathrm{emb}} : \mathcal{W}_\ell(G) \to \mathbb{R}^{\ell \times d}$ as follows:

$$f_{\mathrm{emb}}[i] := h_V(w_i) + \mathrm{proj}(h_{\mathrm{PE}}[i]), \tag{1}$$

where $h_V : V \to \mathbb{R}^d$ is a node embedding function. Here, $h_{\mathrm{PE}}[i]$ serves as an optional position encoding that supplies extra structural context for each node in the walk (Appendix B); when such encoding is employed, it is further processed by the learnable projection mapping $\mathrm{proj} : \mathbb{R}^{d_{\mathrm{PE}}} \to \mathbb{R}^d$. Subsequently, we assume walk embeddings produced by $f_{\mathrm{emb}}$ are processed by a sequence model,

denoted by $f_{\text{seq}} : \mathbb{R}^{\ell \times d} \to \mathbb{R}^{\ell \times d}$. Finally, embeddings from the sequence model are aggregated by a permutation-invariant function. The choice for the function can be simple functions such as taking the mean over random walk representations such as in Wang and Cho [3], Kim et al. [7], or it can be more complex as in Tönshoff et al. [5], Chen et al. [6], which updates a node's representation as the aggregation of its representations across all walks using the aggregation function $f_{\text{agg}} : \mathbb{R}^{m \times \ell \times d} \to \mathbb{R}^{|V| \times d}$:

$$f_{\text{agg}}[w_i] := \frac{1}{N_i(P_m(\mathcal{W}(G)))} \sum_{W \in P_m(\mathcal{W}(G))} \sum_{w_i \in W} f_{\text{seq}}(f_{\text{emb}}(P_m(\mathcal{W}(G))))[i], \qquad (2)$$

where $N_i(P_m(\mathcal{W}(G)))$ represents the number of occurrences of node $i$ in the union of walks in $P_m(\mathcal{W}(G))$. The RWNN layer is defined as the composition $f^l_{\text{RWNN}} = f^l_{\text{agg}} \circ f^l_{\text{seq}}$, while the overall architecture $f_{\text{RWNN}}$ is defined as the stacking of RWNN layers. In the node classification setting, the final node representation $f_{\text{agg}}[i]$ produced by the last RWNN layer is directly utilized for predictions. In graph classification, an additional global pooling function aggregates these node representations into a single representation for the graph.

## 2.4 Random Walk Cover Times

RWNN expressivity depends on how much of the graph its random walks visit (Section 3). Here, we review results on random walk node cover times, $C_V(G)$, the expected number of steps a walk takes to visit all nodes. For a connected graph $G = (V, E)$, the cover time of a general uniform random walk satisfies $C_V(G) = O(|V||E|)$ [22]; in particular, for sparse graphs ($|E| = \Theta(|V|)$) this gives $C_V(G) = O(|V|^2)$. Minimum-degree local rule (MDLR) walks further achieve $C_V(G) = O(|V|^2)$ on all graphs, which is optimal among first-order walks [7, 23]. Non-backtracking walks can also empirically reduce cover times on graphs [5, 6]. Even with these improvements, guaranteeing full node and edge coverage by random walks can require prohibitively long walks or impractically large numbers of walks. We therefore replace walks entirely with *searches* (Section 4), significantly improving on the number of samples required for full coverage in comparison to random walks.

# 3 Expressive Power of Random Walk Neural Networks

In this section, we characterize the expressive power of RWNNs. Our main result establishes that without additional positional or structural encodings, RWNNs with access to the complete multiset of random walks whose lengths scale up to the cover time are exactly as expressive as MPNNs. In practice, however, such assumptions are unrealistic: guaranteeing full node and edge coverage requires walk lengths on the order of the cover time, rendering full coverage computationally infeasible. We then show that in the partial-coverage regime, RWNNs are strictly less expressive than MPNNs. This limitation motivates our random search neural network (RSNN), which achieves full coverage and thus maximal expressivity at significantly lower sampling cost.

## 3.1 The Role of Coverage: RWNNs vs. MPNNs

We first analyze the ideal setting in which the RWNN has access to complete walk sets up to the cover time. In this regime, RWNN expressive power matches that of MPNNs.

**Theorem 3.1** (RWNN-MPNN Equivalence Under Full Coverage (FC)). *Let $G$ be a graph. Let $f^{\text{FC}}_{\text{RWNN}}$ denote an RWNN with injective $f_{\text{seq}}$ and $f_{\text{agg}}$ with no additional positional encodings, applied to the complete multiset of walks $\mathcal{W}_{\leq \ell}(G)$ with lengths up to $\ell = C_E(G)$, the edge cover time of $G$. Let $f_{\text{MPNN}}$ be an MPNN with injective $f_{\text{agg}}$. Then, for all graphs $G, H$,*

$$f_{\text{MPNN}}(G) = f_{\text{MPNN}}(H) \iff f^{\text{FC}}_{\text{RWNN}}(G) = f^{\text{FC}}_{\text{RWNN}}(H).$$

*Hence, $f^{\text{FC}}_{\text{RWNN}} \simeq f_{\text{MPNN}}$ (i.e., $f^{\text{FC}}_{\text{RWNN}}$ and $f_{\text{MPNN}}$ are equal in expressive power).*

Although Theorem 3.1 shows that full-coverage RWNNs and MPNNs are equal in expressivity, RWNNs under full coverage can be more effective empirically. RWNNs leverage expressive sequence models capable of capturing long-range dependencies when given full graph structure in complete sequences. MPNNs instead rely on iterative neighborhood aggregation and are limited in depth by oversmoothing [24] and oversquashing [25], which in practice reduce their expressivity and

capabilities to capture long-range signals. This contrasts our theoretical setup where we assume MPNNs have unlimited depth, allowing them to match full-coverage RWNN expressivity.

Constructing complete walk sets with lengths up to the cover time, however, is typically computationally infeasible. RWNNs can thus fall short of MPNNs under realistic budgets despite their inherent advantages. Indeed, as an immediate consequence of Theorem 3.1, when RWNNs operate under partial coverage, their expressive power is strictly less than that of MPNNs.

**Corollary 3.2** (RWNNs Under Partial Coverage (PC)). *Let $f_{\mathrm{RWNN}}^{\mathrm{PC}}$ denote an RWNN of the same architectural class as in Theorem 3.1 but applied to a multiset of random walks that attains only* partial *node/edge coverage of the input graph. Then, for all graphs $G, H$,*

$$f_{\mathrm{MPNN}}(G) = f_{\mathrm{MPNN}}(H) \implies f_{\mathrm{RWNN}}^{\mathrm{PC}}(G) = f_{\mathrm{RWNN}}^{\mathrm{PC}}(H),$$

*and there exist non-isomorphic graphs $G \not\cong H$ such that*

$$f_{\mathrm{MPNN}}(G) \neq f_{\mathrm{MPNN}}(H) \quad while \quad f_{\mathrm{RWNN}}^{\mathrm{PC}}(G) = f_{\mathrm{RWNN}}^{\mathrm{PC}}(H).$$

*Hence, $f_{\mathrm{RWNN}}^{\mathrm{PC}} \prec f_{\mathrm{MPNN}}$ (partial-coverage RWNNs are strictly less expressive than MPNNs).*

Corollary 3.2 reveals a fundamental limitation of RWNNs: under partial coverage, their expressive power falls below that of classical message passing. Thus, to attain maximal theoretical expressivity, it is essential to design sampling strategies that efficiently guarantee complete coverage. In order to realize the advantages of RWNNs while obtaining maximal expressivity, we introduce RSNNs (Section 4), which replace walks with searches to guarantee full node coverage by construction and achieve full edge coverage with a small number of searches on sparse graphs.

*Insights of the analysis.* In proving Theorem 3.1, we introduce a walk-based color refinement, *Walk Weisfeiler–Lehman* (WWL; Definition A.3), which updates each node using the multiset of walks that visit it. We demonstrate that WWL upper bounds RWNN expressivity (Lemma A.5). Next, we establish that WWL operates on the same object as classical WL: unfolding trees (Definition A.6). We lastly leverage this insight to establish that WWL and WL have equal distinguishing power (Theorem A.9). In essence, this construction *aligns the Weisfeiler–Lehman hierarchy with RWNNs*, unifying the expressive power of two seemingly distinct model classes: RWNNs, which process random walks with sequence models, and MPNNs, which process multisets of node neighborhoods with graph convolution. Formal definitions and details are in Appendix A.

## 4 Random Search Neural Networks (RSNNs)

Motivated by our analysis of RWNNs, we propose a new sampling strategy that efficiently achieves the necessary conditions for maximal expressivity: full node and edge coverage. Since random walks require either prohibitively long walks or an impractically large number of walks to guarantee full coverage, we introduce random search neural networks (RSNNs), which represent graphs as collections of random searches. Notably, a single search guarantees full node coverage, and under the sparse graph assumption, only $O(\log(|V|))$ searches are needed to capture all edges. This significantly reduces the sampling complexity compared to the $O(|V|)$ requirement for traditional RWNNs, assuming walk lengths scale on the order of $O(|V|)$. When paired with a maximally expressive sequence model, RSNNs emerge as universal approximators on graphs. Moreover, we provably show RSNNs are probabilistically invariant to graph isomorphisms. Hence, the predictor obtained by averaging over searches is an isomorphism-invariant graph function. While the computational cost of a full search can be significantly larger than a short random walk, we focus on sparse graphs where search is computationally feasible, addressing the limitations of RWNNs in these classes of graphs.

### 4.1 Search via Random DFS

RSNNs leverage a random depth-first search (DFS) procedure to obtain sequences from an input graph $G$. We utilize a DFS rather than a breadth-first search to better preserve continuity in the sequence. RSNN obtains a random DFS by sampling a uniform root and independent random neighbors at each vertex; this induces a distribution over DFS sequences we denote $\mathcal{S}_{\mathrm{DFS}}(G)$. RSNN collects $m$ independent searches to form the set $P_m(\mathcal{S}_{\mathrm{DFS}}(G)) = \{S_1, \ldots, S_m\}$. Once these searches are obtained, RSNNs leverage all the advances of RWNNs but with new benefits. We apply the recording function (Equation (1)) to each search, including positional encodings from Tönshoff et al. [5] to distinguish between disconnected nodes and true connections in the sequence. Search embeddings are then processed with a sequence model and the node aggregation function (Equation (2)) (Figure 2).

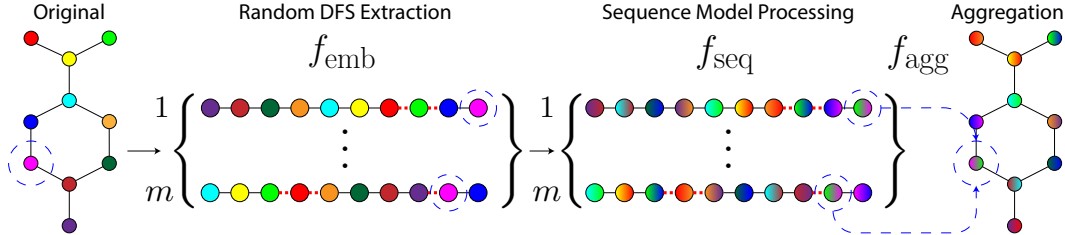

Figure 2: Overview of an RSNN layer. Starting from an input graph, $m$ random depth-first searches are extracted and encoded via $f_{\text{emb}}$. Additional positional encodings indicate discontinuities in the sequence (e.g., ●--● in search 1). These sequences are processed by a sequence model $f_{\text{seq}}$, and final node representations are aggregated across sequences using $f_{\text{agg}}$. We highlight in blue the flow of a selected node representation (shown as ●) as it is tracked through each stage of the RSNN layer.

## 4.2 From Efficient Graph Coverage to Universal Approximation in RSNNs

In this section, we establish the theoretical foundations of RSNNs by demonstrating how our random search strategy efficiently obtains full graph coverage. Central to our analysis is the observation that the subgraphs induced by DFS sequences are spanning trees. Leveraging this insight, we prove the following key lemma showing that for sparse graphs with bounded degree, a logarithmic number of random searches is sufficient to guarantee full node and edge coverage with high probability.

**Lemma 4.1** (Logarithmic Sampling Yields Full Edge Coverage). *Let $G = (V, E)$ be a connected graph with $|E| \leq C|V|$ for some constant $C$ and a bounded maximum degree $d_{\max}$. Let $S_1, S_2, \ldots, S_m$ be $m$ independent random searches sampled from $G$, and let $T_1, T_2, \ldots, T_m$ be their corresponding induced spanning trees. Then, for small $\delta \ll 1$, if*

$$m \geq \frac{\ln\left(\frac{C|V|}{\delta}\right)}{\ln\left(\frac{d_{\max}}{d_{\max}-1}\right)}, \tag{3}$$

*the union of $T_1, T_2, \ldots, T_m$ contains every edge in $E$ with probability at least $1 - \delta$.*

In contrast to RWNNs, which require $m = O(|V|)$ random walks of length $\ell = O(|V|)$, RSNNs achieve complete coverage with $m = O(\log(|V|))$ searches of length $\ell = O(|V|)$. With full node and edge coverage, RSNNs are able to capture all the information necessary to represent any function on graphs. Intuitively, this means that under our sampling strategy, RSNNs are universal approximators: they can approximate any graph function arbitrarily well, provided they are paired with a universal sequence model such as transformers or LSTMs [26, 27].

**Theorem 4.2** (Universal Approximation by RSNNs on Sparse Graphs with Bounded Degree). *Let $\epsilon > 0$ and let $f : \mathcal{G} \to \mathbb{R}^d$ be any continuous graph-level function, where $\mathcal{G}$ is the space of sparse graphs with $|E| = O(|V|)$, maximum degree at most $d_{\max}$, and size at most $|V| \leq n_{\max}$. Assume $m$ satisfies Equation (3), so that full coverage is achieved with probability at least $1 - \delta$. Then, with probability at least $1 - \delta$ there exists an RSNN configuration such that*

$$\|f_{\text{RSNN}}(G) - f(G)\| < \epsilon \quad \text{for all } G \in \mathcal{G}, \tag{4}$$

## 4.3 From Expressivity to Invariance: Isomorphism Invariance of RSNNs

Having established the expressive capabilities of RSNNs, we now turn to invariance. For graphs, the target symmetry is isomorphism invariance: for all $G \cong H$, an isomorphism-invariant graph function satisfies $f(G) = f(H)$. Graph functions that capture the symmetry enjoy learning and generalization benefits. Because RSNNs are randomized functions, we adopt the notion of probabilistic invariance [7, 28]: for all $G \cong H$, the random outputs satisfy $f(G) \overset{d}{=} f(H)$. Intuitively, a randomized graph function is probabilistically invariant to graph isomorphisms if its distribution is unchanged under any graph isomorphism. We demonstrate that the randomized DFS procedure used by RSNNs is probabilistically invariant; consequently, the RSNN predictor $f_{\text{RSNN}}$ is invariant in distribution, and its expectation $\Phi(G) \coloneqq \mathbb{E}[f_{\text{RSNN}}(G)]$ is an isomorphism-invariant function on graphs.

**Theorem 4.3** (Probabilistic Isomorphism-Invariance of RSNN). *A randomized search procedure on a graph $G$ produces a sequence $S^G = (s_0^G, \ldots, s_{|V(G)|}^G)$ of visited vertices. We say the procedure is*

probabilistically invariant *to graph isomorphisms if for all graph isomorphisms $\pi$,*

$$\left(\pi(s_0^G), \ldots, \pi(s_{|V(G)|}^G)\right) \;\overset{d}{=}\; (s_0^H, \ldots, s_{|V(H)|}^H) \quad \text{for all } G \overset{\pi}{\cong} H.$$

*The randomized DFS procedure used in RSNNs satisfies the above definition. Hence, RSNNs satisfy probabilistic invariance: for all $G \cong H$, $f_{\mathrm{RSNN}}(G) \;\overset{d}{=}\; f_{\mathrm{RSNN}}(H)$, and the averaged predictor $\Phi(G) := \mathbb{E}\big[f_{\mathrm{RSNN}}(G)\big]$ is an invariant function on graphs: $\Phi(G) = \Phi(H)$ for all $G \cong H$.*

**Learning the invariance.** In addition to being invariant in expectation, we show that RSNNs can learn the optimal invariant predictor throughout training even under limited sampling budgets, where the expectation is only approximated (e.g., $m = 1$ sampled search for each forward pass in the parameter update). At inference, the invariant predictor can then be computed exactly or estimated by the Monte Carlo estimator. Our result follows Murphy et al. [29, 30]. For RSNN parameters $\mathbf{W}$, define the model output on a graph $G$ and a sampled search set $S \sim \mathcal{S}_{\mathrm{DFS}}(G)$ as $f_{\mathrm{RSNN}}(G, S; \mathbf{W})$.

**Corollary 4.4** (SGD converges to the invariant objective). *Let $\ell(\cdot, y)$ be differentiable and define*

$$L(\mathbf{W}) \;=\; \mathbb{E}_{(G,y)\sim\mathcal{D}}\,\mathbb{E}_{S\sim\mathcal{S}_{\mathrm{DFS}}(G)}\big[\ell\big(f_{\mathrm{RSNN}}(G, S; \mathbf{W}), y\big)\big].$$

*At each step $t$, sample a mini-batch $\mathcal{B}_t = \{(G_t^{(i)}, y_t^{(i)})\}_{i=1}^B$ i.i.d. from $\mathcal{D}$ and, for each $i$, draw a single $S_t^{(i)} \sim \mathcal{S}_{\mathrm{DFS}}(G_t^{(i)})$ independently of $\mathbf{W}_t$; update*

$$\mathbf{W}_{t+1} \;=\; \mathbf{W}_t \;-\; \eta_t \frac{1}{B} \sum_{i=1}^B \nabla_{\mathbf{W}}\, \ell\big(f_{\mathrm{RSNN}}(G_t^{(i)}, S_t^{(i)}; \mathbf{W}_t), y_t^{(i)}\big).$$

*Then $\mathbb{E}\Big[\frac{1}{B}\sum_{i=1}^B \nabla_{\mathbf{W}}\ell\big(f_{\mathrm{RSNN}}(G_t^{(i)}, S_t^{(i)}; \mathbf{W}_t), y_t^{(i)}\big)\Big] = \nabla_{\mathbf{W}}L(\mathbf{W}_t)$, i.e., the mini-batch gradient is an unbiased estimator of $\nabla L(\mathbf{W}_t)$. Under standard SGD conditions, $\mathbf{W}_t$ converges almost surely to an optimizer $\mathbf{W}^\star$ of the invariant objective.*

**Inference.** Given a fixed point $\mathbf{W}^\star$ and a new test graph $G'$, the invariant prediction is $\mathbb{E}_S[f_{\mathrm{RSNN}}(G', S; \mathbf{W}^\star)]$, which can be exactly computed or approximated with the estimator $\frac{1}{m}\sum_{j=1}^m f_{\mathrm{RSNN}}(G', S_j; \mathbf{W}^\star)$ where $S_1, \ldots, S_m \overset{\text{i.i.d.}}{\sim} \mathcal{S}_{\mathrm{DFS}}(G')$.

## 4.4 Runtime Complexity

We compare the sampling costs of RSNNs and RWNNs. In our approach, each random search corresponds to a DFS traversal. Assuming a sparse graph, a single DFS has a worst-case cost of $O(|V|)$, and obtaining $m$ searches requires $O(m|V|)$ time, efficient and computationally feasible in small to medium-sized graphs. In contrast, RWNNs generate $m$ random walks of length $\ell$, with total sampling cost $O(m\ell)$. When $\ell \ll |V|$, random walk sampling can be faster than random search extraction. However, as we have shown, short walks fail to capture global structure, leading to reduced expressivity. Thus, while RSNN sampling is more expensive when $\ell$ is small, its increased coverage and performance can justify its cost, especially in graphs where full structure is critical.

## 5 Experiments & Results

Through empirical evaluation we aim to answer the following research questions, extending our theory by testing RSNNs on datasets with factors not explicitly addressed in the theoretical analysis (e.g., class imbalance, rich node features), and testing RSNNs against models beyond our theory such as canonical approaches (e.g., SMILES, Fingerprints) used commonly in molecular analysis.

- **RQ1 (Discriminative performance):** How does RSNN discriminative performance compare to standard baselines and RWNNs across sparse graph benchmark tasks?
- **RQ2 (Node and edge coverage):** Do RSNNs achieve higher node and edge coverage than RWNNs as the number of sampled searches $m$ increases, and does this increased coverage translate into improved task performance?
- **RQ3 (Generalization to larger and denser graphs):** How do RSNNs perform on larger and denser graphs, where attaining full edge coverage is computationally expensive?

## 5.1 Experimental Setup

**Datasets.** We focus our analysis on molecular and protein benchmarks, domains where RWNNs have demonstrated strong empirical performance and where efficient coverage, long-range dependencies, and high expressivity are essential [5, 7, 31]. Importantly, RSNNs are not intended as a solution across all domains, but as a principled alternative for sparse graphs requiring representations that capture global structure. Specifically, we evaluate on four small-scale molecular graph classification datasets from MoleculeNet [32]: **CLINTOX**, **SIDER**, **TOX21**, and **BBBP**. These benchmarks span diverse molecular tasks such as toxicity and adverse reaction prediction, with graph sizes ranging from tens to hundreds nodes. We also include four protein graph classification datasets from ProteinShake [33]: **EC Subclass**, **EC Mechanism**, **SC Class**, and **SC Family**. Protein graphs are significantly larger than molecules, ranging up to thousands of nodes, making it more difficult to capture global structure. To assess scalability, we evaluate on large-scale molecular benchmarks with hundreds of thousands of graphs from Open Graph Benchmark [34]: **PCBA-1030**, **PCBA-1458**, and **PCBA-4467**. Lastly, to test generalization to larger and denser graphs, we evaluate on **NeuroGraph-task**, a brain graph benchmark, where the task is to predict one of seven mental states (e.g., emotion processing). We provide descriptive statistics for all datasets in Tables 1 and 2.

**Baselines.** First, we compare to standard molecular learning baselines: (1) **SMILES**, a sequence model applied to canonical SMILES [35]; (2) **GCN** [36] and (3) **GIN** [15], message-passing GNNs; and (4) **GT** [12], a graph transformer model. In addition, we compare to (5) **Fingerprint**, a multi-layer perceptron trained on hand-crafted chemical descriptors known to be effective in molecular tasks [37]. Importantly, **SMILES** and **Fingerprint** are not applicable in protein graphs. Second, we consider four RWNN variants as baselines for comparison: (6) **RWNN-base**, which employs uniform random walks of length $\ell$ with mean aggregation over walk representations [4]; (7) **RWNN-anon**, which augments the base model with a node anonymization strategy from Wang and Cho [3]; (8) **RWNN-mdlr**, which uses minimum-degree local rule walks from [7], anonymization, and mean aggregation; (9) **CRAWL** [5], which applies non-backtracking walks with node-level aggregation. We consider three sequence models for $f_{\text{seq}}$: (a) GRU [38], (b) LSTM[26], and (c) transformer [27].

**Training and Evaluation.** To ensure fair comparisons, all RWNNs and RSNN are configured with the same number of samples $m$, and RWNN walk lengths are set to $\ell = |V|$, the number of nodes per graph, so that asymptotic runtimes are equivalent across methods. On molecular benchmarks, we sample a new set of $m$ walks for each forward pass during training, and on protein benchmarks, we precompute the set of $m$ walks before training. Following each dataset's protocol, performance is computed as AUC or accuracy. We report median (min, max) performance over five random splits (60/20/20), which is more robust than mean and standard deviation for small sample sizes. All models are trained on a machine equipped with $8\times$ NVIDIA GeForce GTX 1080 Ti GPUs; if a model does not converge within 24 hours, we omit it from evaluation. All remaining details are in Appendix D[1].

## 5.2 RQ1 & RQ2: Discriminative Performance and Coverage

First, RSNNs significantly outperform standard baselines across all benchmarks, demonstrating their effectiveness for molecular and protein learning (Table 1). Notably, at $m = 16$, RSNNs match or exceed the performance of Fingerprint models, which do not rely on learned representations and instead use features designed by domain experts. For all RWNNs and RSNN, we present results using GRU, which performs best empirically, and include additional results for LSTMs and transformers in the Appendix C, where we observe similar trends. Compared to existing RWNNs, RSNNs exhibit greater expressivity at low sampling budgets; with a single search ($m = 1$), RSNN significantly outperforms all RWNN variants at the same budget. Moreover, across all molecular benchmarks, RSNNs at $m = 1$ match or exceed the best-performing RWNNs at $m = 16$, highlighting their sample efficiency. While performance differences narrow at $m = 16$ on molecular benchmarks, RSNNs retain a substantial lead on larger protein graphs, underscoring their expressivity in structurally complex settings. On large-scale molecular benchmarks, training both RWNNs and RSNNs with $m > 1$ becomes computationally infeasible, exceeding the 24-hour time budget. At $m = 1$, however, RSNNs maintain strong performance and substantially outperform RWNNs (Table 2), demonstrating RSNNs' robustness under sampling constraints when computation is limited.

---

[1]Code can be found at:
https://github.com/MLD3/RandomSearchNNs

Table 1: Median (min, max) of performance across test splits on molecular and protein benchmarks. We highlight in **blue** the best model for each value of $m$. We use "—" to indicate when a method is not applicable (Fingerprint/SMILES) or when training exceeds 24 hours (GT). RSNNs consistently outperform all RWNN variants at $m = 1$. While RWNNs approach RSNN performance on molecular benchmarks at $m = 16$, RSNNs outperform RWNNs across all $m$ on protein benchmarks.

| | | Small Scale Molecular Benchmarks (AUC ↑) | | | | Protein Benchmarks (ACC ↑) | | | |
| --- | --- | --- | --- | --- | --- | --- | --- | --- | --- |
| | | **CLINTOX** | **SIDER** | **BBBP** | **TOX21** | **SC CL** | **SC FAM** | **EC SUB** | **EC MEC** |
| | **# Graphs** | 1.5K | 1.5K | 2K | 8K | 10K | 10K | 15K | 15K |
| | **Avg. $|V|$** | 26.1 | 33.6 | 23.9 | 18.6 | 217.5 | 217.5 | 304.9 | 306.4 |
| | **Avg. $|E|$** | 28.0 | 35.4 | 26.0 | 16.9 | 593.8 | 593.8 | 843.4 | 846.9 |
| | **# Classes** | 2 | 2 | 2 | 2 | 5 | 1000 | 24 | 31 |
| NA | **Fingerprint** | 66.5 (52.3, 74.9) | 70.4 (66.6, 74.5) | 86.2 (83.4, 92.5) | 79.1 (75.1, 81.0) | — | — | — | — |
| | **SMILES** | 62.5 (45.7, 68.6) | 61.5 (57.6, 66.4) | 71.9 (65.5, 75.3) | 71.3 (66.4, 73.8) | — | — | — | — |
| | **GT (full)** | 57.1 (46.5, 73.5) | 64.3 (57.9, 69.0) | 75.8 (62.6, 84.0) | 67.8 (64.8, 73.9) | — | — | — | — |
| | **GCN** | 62.4 (56.9, 74.7) | 64.2 (62.4, 70.3) | 73.9 (68.9, 81.4) | 67.5 (63.1, 71.9) | 63.4 (62.8, 64.9) | 3.9 (1.1, 5.3) | 31.2 (28.0, 33.1) | 52.8 (51.9, 53.1) |
| | **GIN** | 59.7 (54.1, 72.4) | 66.5 (64.0, 69.9) | 75.3 (49.4, 85.3) | 66.9 (64.6, 73.4) | 68.0 (67.9, 69.2) | 10.4 (8.7, 11.7) | 37.2 (33.5, 38.3) | 57.4 (56.1, 59.5) |
| $m = 1$ | **RWNN-base** | 71.0 (54.9, 79.5) | 62.5 (55.9, 67.3) | 74.1 (56.7, 82.8) | 71.5 (68.8, 76.3) | 44.5 (42.9, 45.4) | 2.2 (1.6, 2.8) | 26.7 (24.8, 27.9) | 47.3 (46.1, 48.4) |
| | **RWNN-anon** | 68.2 (52.5, 87.2) | 64.1 (57.0, 67.3) | 74.8 (69.0, 82.6) | 71.2 (69.3, 75.0) | 45.4 (41.5, 45.9) | 4.6 (4.2, 5.8) | 26.9 (26.0, 28.7) | 47.1 (45.6, 48.2) |
| | **RWNN-mdlr** | 70.7 (60.4, 76.1) | 59.8 (57.0, 65.9) | 76.1 (72.1, 81.6) | 70.8 (66.6, 75.3) | 43.3 (42.9, 45.1) | 4.5 (3.7, 4.7) | 26.7 (26.5, 27.2) | 47.2 (46.0, 48.2) |
| | **CRAWL** | 70.0 (64.6, 73.6) | 64.2 (56.1, 67.2) | 77.6 (68.8, 81.5) | 71.7 (66.6, 75.3) | 53.0 (50.7, 53.4) | 5.2 (3.4, 5.8) | 28.7 (27.6, 29.6) | 47.0 (46.2, 47.6) |
| | **RSNN (ours)** | **88.1 (84.9, 91.5)** | **66.2 (63.0, 72.4)** | **87.5 (80.3, 89.9)** | **79.8 (77.2, 83.4)** | **62.2 (60.0, 65.6)** | **13.9 (10.6, 14.9)** | **36.8 (36.5, 38.3)** | **49.8 (48.2, 50.8)** |
| $m = 4$ | **RWNN-base** | 83.6 (76.5, 86.7) | 64.4 (59.9, 71.9) | 84.2 (77.2, 87.0) | 76.3 (71.9, 80.9) | 53.0 (52.5, 54.1) | 3.7 (3.3, 5.4) | 32.7 (32.1, 34.5) | 48.1 (47.1, 48.8) |
| | **RWNN-anon** | 84.7 (80.3, 89.5) | 65.6 (61.5, 68.8) | 82.0 (77.1, 85.4) | 77.2 (73.5, 79.2) | 52.7 (51.7, 53.1) | 6.4 (5.2, 7.5) | 32.9 (31.2, 34.2) | 47.9 (46.5, 50.3) |
| | **RWNN-mdlr** | 82.9 (77.9, 90.4) | 65.5 (60.4, 72.4) | 81.9 (79.2, 88.0) | 76.9 (72.6, 80.2) | 51.5 (50.2, 52.5) | 6.2 (5.4, 7.8) | 32.4 (30.6, 33.6) | 48.2 (47.3, 49.3) |
| | **CRAWL** | 83.0 (76.6, 91.5) | 65.2 (59.5, 71.3) | 84.5 (80.7, 87.0) | 77.6 (75.6, 81.2) | 67.0 (66.6, 67.9) | 10.8 (9.5, 11.4) | 38.2 (37.0, 39.9) | 50.7 (49.9, 51.7) |
| | **RSNN (ours)** | **89.1 (80.9, 91.7)** | **67.0 (61.3, 71.1)** | **88.0 (80.3, 90.5)** | **80.3 (77.3, 84.2)** | **71.7 (70.5, 73.8)** | **15.5 (14.4, 19.2)** | **43.9 (41.7, 44.3)** | **54.8 (51.7, 55.8)** |
| $m = 8$ | **RWNN-base** | 85.0 (82.6, 88.7) | 65.2 (62.8, 70.2) | 84.1 (81.0, 91.1) | 78.3 (72.1, 81.3) | 57.0 (55.5, 58.5) | 6.1 (4.3, 6.9) | 35.5 (34.8, 36.9) | 49.7 (48.2, 52.0) |
| | **RWNN-anon** | 86.6 (81.8, 92.7) | 67.8 (60.3, 70.7) | 83.9 (78.2, 85.3) | 78.9 (76.1, 82.0) | 55.0 (53.5, 58.4) | 9.3 (8.6, 10.0) | 36.2 (35.8, 37.0) | 49.3 (48.8, 50.3) |
| | **RWNN-mdlr** | 83.9 (78.0, 87.5) | 64.9 (61.8, 69.1) | 84.9 (81.5, 86.7) | 77.6 (75.0, 79.0) | 54.9 (52.1, 56.9) | 9.2 (8.4, 10.7) | 35.5 (34.5, 36.7) | 49.6 (48.3, 51.8) |
| | **CRAWL** | 86.5 (83.6, 91.4) | 66.1 (62.1, 69.9) | 86.0 (82.8, 89.6) | 79.1 (76.7, 82.1) | 72.7 (71.7, 73.3) | 14.1 (10.2, 17.6) | 43.7 (43.0, 45.4) | 54.7 (51.6, 55.0) |
| | **RSNN (ours)** | **88.3 (80.1, 91.3)** | **67.6 (63.3, 69.2)** | **88.6 (83.6, 90.3)** | **82.2 (77.3, 85.3)** | **74.4 (74.1, 75.4)** | **16.0 (14.5, 19.2)** | **46.3 (46.0, 49.4)** | **57.1 (56.5, 57.7)** |
| $m = 16$ | **RWNN-base** | 87.8 (84.9, 91.4) | **67.2 (64.6, 71.4)** | 86.0 (83.7, 88.1) | 80.0 (75.6, 81.8) | 59.0 (58.4, 60.2) | 10.9 (9.6, 11.4) | 37.2 (36.1, 39.3) | 51.7 (51.4, 53.0) |
| | **RWNN-anon** | 85.9 (81.7, 91.8) | 66.5 (61.1, 69.3) | 85.8 (80.1, 88.1) | 79.2 (75.9, 82.2) | 60.1 (58.3, 61.5) | 10.2 (8.1, 12.4) | 39.3 (38.5, 40.6) | 51.7 (50.4, 53.2) |
| | **RWNN-mdlr** | 85.9 (81.5, 89.9) | 65.7 (63.5, 70.1) | 85.4 (80.8, 90.5) | 79.1 (77.7, 83.0) | 59.5 (56.7, 61.0) | 11.2 (9.4, 11.7) | 39.1 (38.4, 40.2) | 51.3 (49.9, 51.9) |
| | **CRAWL** | **89.1 (80.5, 91.1)** | 65.3 (61.4, 70.8) | 87.0 (81.7, 90.3) | 80.9 (77.4, 82.6) | 76.2 (73.6, 77.4) | 15.5 (13.6, 16.0) | 48.7 (46.1, 49.3) | 57.4 (56.8, 58.6) |
| | **RSNN (ours)** | 88.5 (82.0, 93.7) | 67.1 (65.0, 74.0) | **89.4 (83.0, 91.7)** | **82.2 (78.0, 84.1)** | **77.0 (75.0, 77.2)** | **19.0 (15.3, 20.1)** | **50.0 (49.5, 52.0)** | **59.5 (57.1, 60.0)** |

We compare how node/edge coverage and performance varies with the number of walks or searches for RSNNs and CRAWL, the strongest RWNN baseline (Figure 3). Across all benchmarks, we observe a strong correlation between coverage and model performance. On molecular graphs, RSNNs achieve full node and high edge coverage with a single search ($m = 1$), resulting in strong initial performance. This aligns with our theoretical analysis: each RSNN search guarantees node coverage by construction, and only a few searches are needed to achieve full edge coverage in sparse graphs. In contrast, CRAWL begins with low node and edge coverage and only reaches RSNN-level performance at $m = 16$, once coverage converges, highlighting RWNN limitations under small sampling budgets. On larger protein graphs, both coverage and performance improve more gradually, but RSNNs retain a consistent performance advantage across all $m$, underscoring the benefit of efficient coverage in larger graphs.

Table 2: Median (min, max) AUC on large scale molecular benchmarks. We highlight in **blue** the best model. RSNNs outperform all RWNNs across all tasks.

| | | Large Scale Molecular Benchmarks (AUC ↑) | | |
| --- | --- | --- | --- | --- |
| | | **PCBA-1030** | **PCBA-1458** | **PCBA-4467** |
| | **# Graphs** | 160K | 195K | 240K |
| | **Avg. $|V|$** | 24.3 | 25.1 | 25.3 |
| | **Avg. $|E|$** | 26.2 | 27.1 | 27.2 |
| $m = 1$ | **RWNN-mdlr** | 63.5 (62.3, 64.3) | 76.2 (75.4, 76.7) | 75.4 (75.4, 76.0) |
| | **CRAWL** | 64.2 (62.5, 64.5) | 77.0 (76.8, 77.2) | 75.6 (75.2, 75.7) |
| | **RSNN** | **78.8 (78.1, 79.3)** | **87.0 (86.7, 87.4)** | **85.2 (84.3, 85.3)** |

## 5.3 RQ3: Generalization to Larger and Denser Graphs

Table 3: Median (min, max) of accuracy on Neuro-Graph benchmark. We highlight in **blue** the best model. RSNNs outperform CRAWL across $m = 4, 16$.

| | **NeuroGraph-task** | | |
| --- | --- | --- | --- |
| **# Graphs** | 7500 | | |
| **Avg. $|V|$** | 400 | | |
| **Avg. $|E|$** | 7029 | | |
| **Max degree** | 153 | | |
| | $m = 1$ | $m = 4$ | $m = 16$ |
| **CRAWL** | **63.4 (59.4, 64.5)** | 77.5 (74.4, 78.9) | 68.3 (30.1, 87.9) |
| **RSNN** | 58.9 (57.1, 61.5) | **80.4 (78.8, 82.6)** | **86.5 (76.5, 88.9)** |

To assess generalization beyond small and sparse regimes, we evaluate on a **NeuroGraph** benchmark of brain graphs. These graphs are substantially larger than molecules and denser than proteins, making full edge coverage expensive for both walks and searches. We compare RSNN against CRAWL. RSNN outperforms CRAWL at $m = 4, 16$, indicating that RSNNs can leverage structure even when full coverage is expensive and that their performance advantage remains on larger and denser graphs.

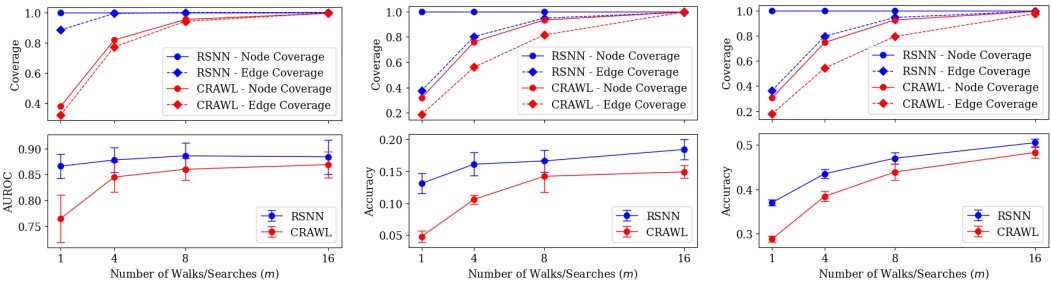

(a) BBBP Molecular Graph     (b) Structural Family Protein Graph    (c) Enzyme Subclass Protein Graph

Figure 3: Coverage vs. performance across benchmarks. RSNNs achieve higher coverage and performance at low sample sizes, while CRAWL only approaches RSNN coverage and performance at $m = 16$, highlighting a strong correlation between coverage and performance.

# 6 Discussion and Conclusion

We present the first theoretical analysis of RWNNs under realistic sampling constraints, showing that their expressivity is fundamentally limited without full node and edge coverage, even in small graphs. We prove that under partial coverage, RWNNs are strictly less expressive than traditional MPNNs. To address this, we introduce RSNNs, which use random depth-first search to guarantee full node coverage and edge coverage with only a logarithmic number of samples in sparse graphs. When paired with expressive sequence models, we show that RSNNs are universal approximators. Furthermore, RSNNs are also probabilistically invariant to graph isomorphisms. Empirically, RSNNs consistently outperform RWNNs on both molecular and protein benchmarks, requiring up to $16\times$ fewer samples to achieve comparable performance.

Our work builds on recent work in RWNNs that combines random walks with expressive sequence models [3–8]. These works explore various walk strategies, including uniform walks [3, 4], non-backtracking walks [5, 6], minimum-degree local rule walks [7], and learnable walks [8], and propose architectural improvements to enhance expressivity and performance. We critically examine the expressivity of RWNNs under realistic sampling constraints, relaxing prior assumptions that walks are as long as cover times. Based on our analysis, we propose to replace random walks entirely with random searches, leading to RSNNs, a more sample-efficient and expressive alternative.

Our work is not without limitations. In particular, RSNNs are tailored to sparse, medium-sized graphs. How to scale RSNN to extremely large, densely connected graphs remains an open question. In such settings, full-depth searches may become prohibitively expensive, and edge coverage may scale less efficiently. A promising direction is to explore truncated searches that capture key structural signal while reducing computation. This raises new questions about how coverage and expressivity behave under partial searches, particularly in dense regimes where full coverage is infeasible.

Despite the focused scope, our results are promising: RSNNs match or exceed RWNN performance with significantly fewer samples and maintain a clear advantage across benchmarks. These findings underscore the value of replacing random walk sampling with search-based sampling in graph learning. More broadly, this work highlights the importance of moving beyond local neighborhoods toward sampling strategies that capture global structure. By leveraging efficient coverage through random searches, RSNNs offer a principled, expressive, and sample-efficient framework for learning on sparse graphs, laying the foundation for future exploration in other settings.

**Acknowledgements**

This material is based upon work supported by the U.S. Department of Energy, Office of Science, Office of Advanced Scientific Computing Research, Department of Energy Computational Science Graduate Fellowship under Award Number DE-SC0023112. It was also partially supported by National Science Foundation under Grants No. IIS 2212143 and IIS 2504090. We thank the anonymous reviewers and members of the MLD3 lab for their valuable feedback.

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

# A  Mathematical Proofs

## A.1  Random Walk Neural Network Expressive Power

We begin with the 1–Weisfeiler–Lehman (WL) color refinement, which iteratively updates each node's color by hashing its current color together with the multiset of its neighbors' colors (Section A.1.1). WL is known to upper-bound MPNN expressivity [15, 39]. We then introduce *Walk Weisfeiler–Lehman* (WWL), a walk-based refinement that updates a node from the multiset of *colored walks* of length $\leq \ell$ that terminates at it (Section A.1.2). We establish monotonicity of WWL in the number of refinement rounds $t$, the maximum walk length $\ell$, and the initialization $\pi^{(0)}$ (i.e., richer initial features yield finer partitions). We further show that WWL upper-bounds the expressive power of RWNNs (without positional/ID signals). Finally, using *unfolding trees*, which simultaneously captures the nodes visible to $t$ rounds of message passing and encodes all root-terminating walks of length $\leq t$, we prove the main expressivity results that unify MPNNs and RWNNs: equivalence under full coverage and strict separation under partial coverage (Section A.1.3). We provide a more detailed review of existing WL variants and their relation to WWL in Appendix E.

### A.1.1  Weisfeiler–Lehman (WL)

We begin with the 1-dimensional Weisfeiler–Lehman (WL) color-refinement procedure, which upper-bounds the expressive power of message-passing GNNs and, as we will show later, also upper-bounds RWNN expressive power. Intuitively, WL iteratively refines node labels by hashing each node's current label together with the multiset of its neighbors' labels.

**Definition A.1** (1-WL color refinement). Let $G = (V, E)$ be an unlabeled graph and let $\mathcal{N}(u)$ denote the neighbors of $u \in V$. Initialize a coloring $\pi_{\mathrm{WL}}^{(0)} : V \to \Sigma$ with a constant value (e.g., $\pi_{\mathrm{WL}}^{(0)}(u) = 1$ for all $u$). For $t \geq 0$, update

$$\pi_{\mathrm{WL}}^{(t+1)}(u) \;=\; \mathrm{Hash}\Big( \pi_{\mathrm{WL}}^{(t)}(u), \; \big\{\!\!\big\{ \, \pi_{\mathrm{WL}}^{(t)}(v) \, : \, v \in \mathcal{N}(u) \, \big\}\!\!\big\} \Big) \quad \forall u \in V,$$

where $\mathrm{Hash}$ is injective and maps pairs of the form (current color, neighbor colors) to $\Sigma$. The process *stabilizes* at the first $t^\star$ for which $\pi_{\mathrm{WL}}^{(t^\star)} = \pi_{\mathrm{WL}}^{(t^\star+1)}$; we denote the stable coloring by $\pi_{\mathrm{WL}}^{(\infty)}$.

To compare two graphs $G$ and $H$, run 1-WL on each. If the stable color multisets differ (e.g., some color has a different node count), the graphs are certified non-isomorphic. If the stable colorings agree, the test is inconclusive (the graphs may still be non-isomorphic). For the remainder of the analysis, we write $\alpha \preceq \beta$ for *node-level* colorings $\alpha, \beta : V \to \Sigma$ to mean that $\beta$ refines $\alpha$: if $\beta(u) = \beta(v)$ then $\alpha(u) = \alpha(v)$. These notions coincide with graph-level distinguishability: applying an injective readout on multiset colors $\alpha$ and $\beta$ for $\alpha \preceq \beta$ yields graph-level functions $f_\alpha$ and $f_\beta$ such that $f_\alpha \preceq f_\beta$. Thus, distinguishability at the node-level translates to distinguishability at the graph-level.

WL has been used to quantify the expressive power of MPNNs. For any standard message-passing NN, its expressive power is no greater than that of WL. Moreover, if its aggregation function is injective on the multisets of node neighbors, its expressive power matches that of WL.

**Lemma A.2** (MPNN vs. 1-WL Expressivity [15, 39]). *Let $f_{\mathrm{MPNN}}$ be a MPNN with a permutation-invariant readout, and let $\pi_{\mathrm{WL}}$ denote the 1-WL coloring. Then*

$$f_{\mathrm{MPNN}} \;\preceq\; \pi_{\mathrm{WL}}.$$

*Moreover, if the multiset aggregator $f_{\mathrm{agg}}$ used by $f_{\mathrm{MPNN}}$ is injective, then*

$$f_{\mathrm{MPNN}} \;\simeq\; \pi_{\mathrm{WL}}.$$

### A.1.2  Walk Weisfeiler–Lehman (WWL)

Building on WL, we now align Weisfeiler–Lehman to random walk models by defining a node-level WWL scheme that refines a node's label from the multiset of *colored walks* incident to it.

**Definition A.3** (WWL at length $\ell$). Let $G = (V, E)$ be a graph and $\ell \in \mathbb{N}$. For $L \geq 1$, let

$$\mathcal{W}_L \;=\; \{\, W = (w_0, \ldots, w_L) \in V^{L+1} \, : \, (w_{i-1}, w_i) \in E \; \forall i \in [L] \,\}$$

be the set of length-$L$ walks, and write $\mathcal{W}_{\leq \ell} = \bigcup_{\ell=1}^{\ell} \mathcal{W}_L$, the union of all walks of length $\leq \ell$. For a node $u \in V$, define its terminating-walk neighborhood

$$\mathcal{W}_{\leq \ell}(u) \;=\; \{\, W = (w_0, \ldots, w_L) \in \mathcal{W}_{\leq \ell} \;:\; w_L = u \,\}.$$

Given an initial coloring $\pi^{(0)} : V \to \Sigma$ (e.g., uniform or from node features), define for any walk $\mathrm{col}^{(t)}_{\mathrm{WWL}^\ell}(W) \;=\; \big(\pi^{(t)}_{\mathrm{WWL}^\ell}(w_0), \ldots, \pi^{(t)}_{\mathrm{WWL}^\ell}(w_L)\big)$, the colored walk obtained by applying $\pi^{(t)}_{\mathrm{WWL}^\ell}$ to each node in the walk. The $\mathrm{WWL}^\ell$ update is, for all $u \in V$,

$$\pi^{(t+1)}_{\mathrm{WWL}^\ell}(u) \;=\; \mathrm{Hash}\Big(\pi^{(t)}_{\mathrm{WWL}^\ell}(u), \; \{\!\{\, \mathrm{col}^{(t)}_{\mathrm{WWL}^\ell}(W) \;:\; W \in \mathcal{W}_{\leq \ell}(u) \,\}\!\}\Big).$$

**Lemma A.4** (Monotonicity in $t$, $\ell$, and $\pi_0$)**.** *Fix $\ell \leq \ell'$, $t \leq t'$, and initial colorings $\pi_0 \preceq \pi_0'$. Then*

$$(\text{time}) \quad \pi^{(t)}_{\mathrm{WWL}^\ell(\pi_0)} \;\preceq\; \pi^{(t')}_{\mathrm{WWL}^\ell(\pi_0)}, \tag{5}$$

$$(\text{length}) \quad \pi^{(t)}_{\mathrm{WWL}^\ell(\pi_0)} \;\preceq\; \pi^{(t)}_{\mathrm{WWL}^{\ell'}(\pi_0)}, \tag{6}$$

$$(\text{initialization}) \quad \pi^{(t)}_{\mathrm{WWL}^\ell(\pi_0)} \;\preceq\; \pi^{(t)}_{\mathrm{WWL}^\ell(\pi_0')}. \tag{7}$$

*Consequently, combining each result yields $\pi^{(t)}_{\mathrm{WWL}^\ell(\pi_0)} \preceq \pi^{(t')}_{\mathrm{WWL}^{\ell'}(\pi_0')}$ for $t \leq t'$, $\ell \leq \ell'$, $\pi_0 \preceq \pi_0'$.*

*Proof. Monotonicity in $t$.* At each step, $\pi^{(t+1)}_{\mathrm{WWL}^\ell}(u) = \mathrm{Hash}\big(\pi^{(t)}_{\mathrm{WWL}^\ell}(u), \,\cdot\,\big)$ includes the current color as an input. By injectivity of Hash, if two nodes receive the same new color then they had the same current color. Thus $\pi^{(t)}_{\mathrm{WWL}^\ell} \preceq \pi^{(t+1)}_{\mathrm{WWL}^\ell}$, and induction gives the stated inequality for $t \leq t'$.

*Monotonicity in $\ell$.* Let $\ell \leq \ell'$. For each node $u$, the multiset of colored terminating walks of lengths $\leq \ell$ is obtained from the corresponding multiset for lengths $\leq \ell'$ by the projection that discards all walks of length $> \ell$. Therefore, if two nodes are equal under $\mathrm{WWL}^{\ell'}$, they are equal under $\mathrm{WWL}^\ell$ as well. Injectivity of Hash yields $\pi^{(t)}_{\mathrm{WWL}^\ell} \preceq \pi^{(t)}_{\mathrm{WWL}^{\ell'}}$.

*Monotonicity in the initialization $\pi_0$.* Assume $\pi_0 \preceq \pi_0'$. Then there exists a color-forgetting map $\rho$ with $\pi_0 = \rho \circ \pi_0'$. Apply $\rho$ pointwise to every color in each colored walk: for any terminating walk $W = (w_0, \ldots, w_L)$,

$$(\pi_0(w_0), \ldots, \pi_0(w_L)) = \big(\rho(\pi_0'(w_0)), \ldots, \rho(\pi_0'(w_L))\big).$$

Hence the multiset of $\pi_0$-colored walks at any node is the image, under this deterministic transformation, of the multiset of $\pi_0'$-colored walks. Consequently, equality of the $\pi_0'$-based walk multisets implies equality of the $\pi_0$-based walk multisets, and injectivity of Hash gives $\pi^{(1)}_{\mathrm{WWL}^\ell(\pi_0)} \preceq \pi^{(1)}_{\mathrm{WWL}^\ell(\pi_0')}$. The same argument iterates, since each WWL round recomputes colors from the previous round's coloring via the same construction, yielding $\pi^{(t)}_{\mathrm{WWL}^\ell(\pi_0)} \preceq \pi^{(t)}_{\mathrm{WWL}^\ell(\pi_0')}$ for all $t$. $\qquad\square$

The following lemma is an analogue to expressive results on MPNNs and 1-WL. Intuitively, $\mathrm{WWL}^\ell$ upper bounds RWNN expressivity, and RWNNs can match $\mathrm{WWL}^\ell$ if their aggregator is injective.

**Lemma A.5** (RWNN vs. $\mathrm{WWL}^\ell$ Expressivity)**.** *Let $f^\ell_{\mathrm{RWNN}}$ be a random walk neural network that, for each node $u$, aggregates over the multiset of all terminating walks of lengths $1, \ldots, \ell$ ending at $u$, via a permutation-invariant aggregator and a sequence encoder applied to each walk. Let $\pi_{\mathrm{WWL}^\ell}$ denote the $\mathrm{WWL}^\ell$ coloring.*

1. *(Upper bound) For any choice of encoders/aggregators,*

$$f^\ell_{\mathrm{RWNN}} \;\preceq\; \pi_{\mathrm{WWL}^\ell}.$$

   *That is, if two graphs are indistinguishable by $\mathrm{WWL}^\ell$, they are indistinguishable by $f^\ell_{\mathrm{RWNN}}$.*

2. *(Tightness under injectivity) Suppose the sequence encoder $f_{\mathrm{seq}}$ is injective on length-aware color sequences and the nodewise multiset aggregator $f_{\mathrm{agg}}$ is injective. Then*

$$f^\ell_{\mathrm{RWNN}} \;\simeq\; \pi_{\mathrm{WWL}^\ell}.$$

*Proof. (Upper bound).* We prove by induction on $t$ that $\pi_{\text{WWL}^\ell}^{(t)}(u) = \pi_{\text{WWL}^\ell}^{(t)}(v)$ implies $f_{\text{RWNN}}^{\ell,t}(u) = f_{\text{RWNN}}^{\ell,t}(v)$.

*Base case $t = 0$.* Both procedures start from the same initialization (e.g., uniform or fixed features), so the claim holds trivially.

*Inductive step.* Assume the claim holds at depth $t$. Take $u, v$ with $\pi_{\text{WWL}^\ell}^{(t+1)}(u) = \pi_{\text{WWL}^\ell}^{(t+1)}(v)$. By injectivity of the WWL hash, the entire inputs to the hash coincide, hence

$$\{\!\{\pi_{\text{WWL}^\ell}^{(t)}(W) : W \in \mathcal{W}_{\leq\ell}(u)\}\!\} = \{\!\{\pi_{\text{WWL}^\ell}^{(t)}(W') : W' \in \mathcal{W}_{\leq\ell}(v)\}\!\},$$

where each $\pi_{\text{WWL}^\ell}^{(t)}(W)$ is the length-aware color sequence along the walk $W$. Thus there is a bijection between terminating walks at $u$ and $v$ that preserves these sequences. By the induction hypothesis, matched nodes with equal WWL color at round $t$ have equal RWNN representations at depth $t$. Therefore, for each matched walk pair, the inputs to the per-walk sequence encoder $f_{\text{seq}}$ agree elementwise, so per-walk encodings match; applying the same permutation-invariant multiset aggregator $f_{\text{agg}}$ yields $f_{\text{RWNN}}^{\ell,t+1}(u) = f_{\text{RWNN}}^{\ell,t+1}(v)$. This completes the induction and the upper bound.

*(Equivalence under injectivity).* Assume $f_{\text{seq}}$ is injective on length-aware sequences and $f_{\text{agg}}$ is injective on multisets. Let $u, v$ satisfy $\pi_{\text{WWL}^\ell}^{(t+1)}(u) \neq \pi_{\text{WWL}^\ell}^{(t+1)}(v)$. By injectivity of the WWL hash, either their current colors at round $t$ differ, or their multisets $\{\!\{\pi_{\text{WWL}^\ell}^{(t)}(W) : W \in \mathcal{W}_{\leq\ell}(\cdot)\}\!\}$ differ. In the first case, including (an injective transform of) the current node state in the RWNN update separates $u$ and $v$. In the second case, there is no bijection between the two multisets of colored sequences; since $f_{\text{seq}}$ is injective on sequences and $f_{\text{agg}}$ is injective on multisets, the aggregated RWNN representations must differ at round $t+1$. Combining with the upper bound, we conclude $f_{\text{RWNN}}^{\ell,t} \simeq \pi_{\text{WWL}^\ell}^{(t)}$ for all $t$. $\qquad\square$

### A.1.3   RWNN-MPNN Equivalence Under Full Coverage (Theorem 3.1, Corollary 3.2)

**Unfolding Trees.**   We introduce the *unfolding tree* from Morris et al. [40], Kriege [41], which makes explicit the bridge between Weisfeiler–Lehman (WL) refinement and random walks. For a node $u$ in $G$, the unfolding tree at depth $\ell$ enumerates, with multiplicities, all vertices seen by successive layers of message passing around $u$. Equivalently, every leaf-to-root path in the unfolding tree corresponds to a walk in $G$ that *terminates* at $u$. Hence the unfolding tree simultaneously encodes (i) all messages propagated in a message-passing view and (ii) all terminating walks of length $\leq \ell$ at $u$. We will leverage this structure to relate the expressive power of WL and WWL.

**Definition A.6** (Unfolding tree [40, 41]). Let $G = (V, E)$ be a graph, $\ell \in \mathbb{N}$, and $u \in V$. The *unfolding tree* of depth $\ell$ rooted at $u$, denoted $T^G[\ell, u]$, is the rooted tree defined recursively as follows:

- $T^G[0, u]$ consists of a single root node labeled by $u$.

- For $\ell \geq 1$, $T^G[\ell, u]$ has root labeled by $u$; for each neighbor $v \in \mathcal{N}_G(u)$, attach as a child a fresh copy of $T^G[\ell - 1, v]$.

The first key fact ties WL colors to unfolding trees: WL's $\ell$-round color of a node is exactly the isomorphism type of its depth-$\ell$ unfolding tree.

**Lemma A.7** (WL $\leftrightarrow$ unfolding tree [41]). *Let $G, H$ be graphs, $u \in V(G)$, $v \in V(H)$, and $\ell \geq 1$. Then*

$$\pi_{\text{WL}}^{(\ell)}(u) = \pi_{\text{WL}}^{(\ell)}(v) \quad \Longleftrightarrow \quad T^G[\ell, u] \cong T^H[\ell, v],$$

Unfolding trees also capture terminating walks: every leaf-to-root path in $T^G[\ell, u]$ reads off a unique length-$\ell$ walk in $G$ ending at $u$, and conversely. Let

$$W\big(T^G[\ell, u]\big) = \{\!\{(x_0, \ldots, x_\ell) : (x_0, \ldots, x_\ell) \text{ is a leaf-to-root path in } T^G[\ell, u]\}\!\}$$

be the multiset of vertex-sequences read along leaf-to-root paths (ordered from leaf to root). Let $\mathcal{W}_\ell(u)$ denote the multiset of all length-$\ell$ walks in $G$ that terminate at $u$ (with multiplicity). Then:

**Lemma A.8** (Leaf-to-root paths $\leftrightarrow$ terminating walks [41]). *For any $u \in V(G)$ and $\ell \geq 1$,*

$$W\big(T^G[\ell, u]\big) = \mathcal{W}_\ell(u).$$

**Theorem A.9.** *For any graphs $G, H$ and any $\ell \geq 1$, $\mathrm{WWL}^\ell$ test has exactly the same distinguishing power as the classical 1-dimensional Weisfeiler–Lehman test. Formally,*

$$\pi^{(\infty)}_{\mathrm{WWL}^\ell} \;\simeq\; \pi^{(\infty)}_{\mathrm{WL}}.$$

*Proof.* $\pi^{(\infty)}_{\mathrm{WL}} \preceq \pi^{(\infty)}_{\mathrm{WWL}^\ell}$. For $\ell = 1$, the set of terminating length-1 walks at a node $u$ is exactly its neighbor set $\mathcal{N}(u)$. Hence the $\mathrm{WWL}^1$ update coincides with the WL update, and for every round $t$

$$\pi^{(t)}_{\mathrm{WWL}^1} \;=\; \pi^{(t)}_{\mathrm{WL}}, \quad \text{in particular} \quad \pi^{(\infty)}_{\mathrm{WWL}^1} \;=\; \pi^{(\infty)}_{\mathrm{WL}}.$$

By Lemma A.4 (monotonicity in $\ell$), $\pi^{(t)}_{\mathrm{WWL}^1} \preceq \pi^{(t)}_{\mathrm{WWL}^\ell}$ for all $\ell \geq 1$ and all $t$. Passing to the limit,

$$\pi^{(\infty)}_{\mathrm{WL}} \;=\; \pi^{(\infty)}_{\mathrm{WWL}^1} \;\preceq\; \pi^{(\infty)}_{\mathrm{WWL}^\ell}.$$

$\pi^{(\infty)}_{\mathrm{WWL}^\ell} \preceq \pi^{(\infty)}_{\mathrm{WL}}$. Initializing WWL with the WL limit, $\pi^{(0)} = \pi^{(\infty)}_{\mathrm{WL}}$, it suffices to show that one WWL update makes no further splits. Fix $u \in V(G)$ and $v \in V(H)$ with $\pi^{(\infty)}_{\mathrm{WL}}(u) = \pi^{(\infty)}_{\mathrm{WL}}(v)$. By Lemma A.7, there is a root-preserving isomorphism $\sigma : T^G[\ell, u] \xrightarrow{\cong} T^H[\ell, v]$. By Lemma A.8, leaf-to-root paths in these depth-$\ell$ trees biject with the terminating walks of lengths $1, \ldots, \ell$ at $u$ and $v$, respectively; $\sigma$ also preserves $\mathrm{WL}^\infty$ colors at every node of two unfolding trees. To show this, suppose for contradiction, that there exists $x \in T_G[\ell, u]$ with $\pi^{(\infty)}_{\text{1-WL}}(x) \neq \pi^{(\infty)}_{\text{1-WL}}(\sigma(x))$. Since 1-WL stabilizes in finite time on the finite disjoint union $G \uplus H$, there exists a finite witness round $k^\star \in \mathbb{N}$ such that $\pi^{(k^\star)}_{\text{1-WL}}(x) \neq \pi^{(k^\star)}_{\text{1-WL}}(\sigma(x))$. Let $d$ be the distance from $x$ to the root $u$ in $T_G[\ell, u]$. By the 1-WL update rule, a mismatch at a node at round $k^\star$ forces a mismatch at its parent at round $k^\star + 1$ (the multiset of child colors differs), and inductively a mismatch at the root after $d$ further rounds:

$$\pi^{(k^\star + d)}_{\text{1-WL}}(u) \;\neq\; \pi^{(k^\star + d)}_{\text{1-WL}}(v).$$

This contradicts $\pi^{(\infty)}_{\text{1-WL}}(u) = \pi^{(\infty)}_{\text{1-WL}}(v)$. Hence $\sigma$ must preserve 1-WL colors at every node. Consequently, the leaf-to-root paths in $T_G[\ell, u]$ and $T_H[\ell, v]$ correspond bijectively under $\sigma$ with identical colored sequences. Thus, the corresponding multisets of *$WL^\infty$-colored* terminating-walk sequences at $u$ and $v$ coincide. Together with $\pi^{(0)}(u) = \pi^{(0)}(v)$, the entire inputs to the WWL hash agree at $u$ and $v$, so by injectivity of Hash we obtain $\pi^{(1)}_{\mathrm{WWL}^\ell(\pi^{(\infty)}_{\mathrm{WL}})}(u) = \pi^{(1)}_{\mathrm{WWL}^\ell(\pi^{(\infty)}_{\mathrm{WL}})}(v)$. Hence $\pi^{(\infty)}_{\mathrm{WL}}$ is a fixed point of WWL. By Lemma A.4, WWL is monotone in $t$ and $\pi_0$; since uniform $\preceq \pi^{(\infty)}_{\mathrm{WL}}$, it follows that $\pi^{(\infty)}_{\mathrm{WWL}^\ell} \preceq \pi^{(\infty)}_{\mathrm{WL}}$. Combined with $\pi^{(\infty)}_{\mathrm{WL}} = \pi^{(\infty)}_{\mathrm{WWL}^1} \preceq \pi^{(\infty)}_{\mathrm{WWL}^\ell}$, we conclude $\pi^{(\infty)}_{\mathrm{WWL}^\ell} \simeq \pi^{(\infty)}_{\mathrm{WL}}$. $\qquad\square$

We now leverage the preceding results to prove the main expressivity statements.

**Theorem A.10** (RWNN-MPNN equivalence under full coverage). *Let $G$ be a graph. Let $f^{\mathrm{FC}}_{\mathrm{RWNN}}$ denote an RWNN with injective $f_{\mathrm{seq}}$ and $f_{\mathrm{agg}}$ with no additional positional encodings, applied to the complete multiset of walks $\mathcal{W}_{\leq \ell}(G)$ with lengths up to $\ell = C_E(G)$, the edge cover time of $G$. Let $f_{\mathrm{MPNN}}$ be an MPNN with injective $f_{\mathrm{agg}}$. Then, for all graphs $G, H$,*

$$f_{\mathrm{MPNN}}(G) = f_{\mathrm{MPNN}}(H) \;\iff\; f^{\mathrm{FC}}_{\mathrm{RWNN}}(G) = f^{\mathrm{FC}}_{\mathrm{RWNN}}(H).$$

*Hence, $f^{\mathrm{FC}}_{\mathrm{RWNN}} \simeq f_{\mathrm{MPNN}}$ (i.e., $f^{\mathrm{FC}}_{\mathrm{RWNN}}$ and $f_{\mathrm{MPNN}}$ are equal in expressive power).*

*Proof.* By the standard 1-WL result for message passing (Theorem A.2), an MPNN with injective aggregation satisfies $f_{\mathrm{MPNN}} \simeq \pi_{\mathrm{WL}}$. By the RWNN/WWL correspondence (Lemma A.5), a full-coverage RWNN with injective $f_{\mathrm{seq}}$ and $f_{\mathrm{agg}}$ satisfies $f^{\mathrm{FC}}_{\mathrm{RWNN}} \simeq \pi_{\mathrm{WWL}^\ell}$. Finally, by the equivalence $\pi_{\mathrm{WWL}^\ell} \simeq \pi_{\mathrm{WL}}$ (Theorem A.9), we conclude $f^{\mathrm{FC}}_{\mathrm{RWNN}} \simeq f_{\mathrm{MPNN}}$. $\qquad\square$

**Corollary A.11** (RWNNs under partial coverage). *Let $f^{\mathrm{PC}}_{\mathrm{RWNN}}$ be an RWNN of the same architectural class as in Theorem 3.1, but applied to a multiset of terminating walks that achieves only* partial *node/edge coverage of the input. Then, for all graphs $G, H$,*

$$f_{\mathrm{MPNN}}(G) = f_{\mathrm{MPNN}}(H) \;\implies\; f^{\mathrm{PC}}_{\mathrm{RWNN}}(G) = f^{\mathrm{PC}}_{\mathrm{RWNN}}(H),$$

*and there exist non-isomorphic graphs $G \not\cong H$ such that*

$$f_{\mathrm{MPNN}}(G) \neq f_{\mathrm{MPNN}}(H) \quad \text{while} \quad f_{\mathrm{RWNN}}^{\mathrm{PC}}(G) = f_{\mathrm{RWNN}}^{\mathrm{PC}}(H).$$

*Hence $f_{\mathrm{RWNN}}^{\mathrm{PC}} \prec f_{\mathrm{MPNN}}$.*

*Proof.* Coverage monotonicity (a direct consequence of injectivity and permutation invariance of the aggregator on multisets) implies that removing walks cannot increase distinguishing power, i.e., $f_{\mathrm{RWNN}}^{\mathrm{PC}} \preceq f_{\mathrm{RWNN}}^{\mathrm{FC}} \simeq f_{\mathrm{MPNN}}$, which yields the implication $f_{\mathrm{MPNN}}(G) = f_{\mathrm{MPNN}}(H) \Rightarrow f_{\mathrm{RWNN}}^{\mathrm{PC}}(G) = f_{\mathrm{RWNN}}^{\mathrm{PC}}(H)$. For strictness, start from two isomorphic graphs and form $G$ by adding one isolated vertex and $H$ by adding one pendant vertex (a new vertex attached to an existing node). Then 1-WL (hence an MPNN) distinguishes $G$ and $H$. However, if $f_{\mathrm{RWNN}}^{\mathrm{PC}}$ is applied to walk multisets that exclude all walks visiting the added vertex in both graphs, the remaining covered walks coincide, so $f_{\mathrm{RWNN}}^{\mathrm{PC}}(G) = f_{\mathrm{RWNN}}^{\mathrm{PC}}(H)$. Thus $f_{\mathrm{RWNN}}^{\mathrm{PC}} \prec f_{\mathrm{MPNN}}$. $\qquad\square$

## A.2 Random Search Neural Network Expressive Power (Lemma 4.1, Theorem 4.2)

We first establish a coverage lemma: for any edge $e = \{u, v\}$ in a connected graph $G$ with maximum degree $d_{\max}$, a randomized DFS (uniform start; i.i.d. tie-breaking) includes $e$ in its spanning tree with probability at least $1/d_{\max}$, i.e., $\Pr\big[e \in T_{\mathrm{DFS}}(G)\big] \geq 1/d_{\max}$. Building on this, we show that sampling $O\big(d_{\max} \log |E|\big)$ independent DFS trees suffices to achieve full *edge* coverage with high probability; in bounded-degree sparse graphs ($d_{\max} = O(1)$ and $|E| = \Theta(|V|)$), this reduces to $O(\log |V|)$ searches. Equipped with such full coverage, standard universal components, and shared anonymous integer tags, RSNNs are universal approximators on graphs in the specified family.

**Lemma A.12** (Edge inclusion probability under random DFS). *Consider the following random–DFS procedure on a graph $G$: fix a uniform distribution over the root vertex; independently for each vertex $x$, draw a uniformly random permutation $\pi_x$ of its neighbors; run depth-first search that, upon first visiting $x$, explores neighbors in the order $\pi_x$. Let $T_{\mathrm{DFS}}$ be the resulting DFS spanning tree. For an edge $e = (u, v)$, define*

$$S_u(e) := \big\{ w \in \mathcal{N}(u) \setminus \{v\} \ : \ \text{there exists a } u \to v \text{ path in } G \setminus \{e\} \text{ whose first edge is } (u, w) \big\},$$

*and set $\tau_u(e) := |S_u(e)|$; define $S_v(e)$ and $\tau_v(e)$ analogously. Then*

$$\Pr\big[e \in E(T_{\mathrm{DFS}})\big] \ \geq \ \min\Big\{ \frac{1}{\tau_u(e) + 1}, \ \frac{1}{\tau_v(e) + 1} \Big\} \ \geq \ \frac{1}{\max\{\deg(u), \deg(v)\}} \ \geq \ \frac{1}{d_{\max}}.$$

*Proof.* Let $A$ be the event that $u$ is discovered by DFS before $v$. On $A$, when $u$ is first processed, $v$ is unvisited. The edge $(u, v)$ will be taken as a tree edge iff, in the random neighbor order $\pi_u$, the vertex $v$ appears *before all* neighbors $S_u(e)$ that can lead from $u$ to $v$ without using $e$. The positions of the other neighbors of $u$ are irrelevant: exploring any neighbor not on a path to $v$ first cannot reach $v$ before DFS returns to $u$. Since $\pi_u$ is a uniform permutation, the probability of this sufficient event is exactly $1/(\tau_u(e) + 1)$. A symmetric argument on $A^c$ (i.e., when $v$ is discovered before $u$) gives the bound $1/(\tau_v(e) + 1)$. Unconditionally,

$$\Pr\big[e \in T\big] = \Pr(A) \Pr\big[e \in T \mid A\big] + \Pr(A^c) \Pr\big[e \in T \mid A^c\big] \ \geq \ \min\Big\{ \frac{1}{\tau_u(e) + 1}, \ \frac{1}{\tau_v(e) + 1} \Big\}.$$

where $T$ is a random DFS tree. Finally, $\tau_u(e) \leq \deg(u) - 1$ and $\tau_v(e) \leq \deg(v) - 1$, hence $\min\{1/(\tau_u + 1), 1/(\tau_v + 1)\} \geq 1/\max\{\deg(u), \deg(v)\} \geq 1/d_{\max}$. $\qquad\square$

**Lemma A.13** (Logarithmic Sampling Yields Full Edge Coverage). *Let $G = (V, E)$ be a connected, unweighted graph with $|E| \leq C|V|$ for some constant $C$ and a bounded maximum degree $d_{\max}$. Let $S_1, S_2, \ldots, S_m$ be $m$ independent random searches sampled from $G$, and let $T_1, T_2, \ldots, T_m$ be their corresponding induced spanning trees. Then, for small $\delta \ll 1$, if*

$$m \geq \frac{\ln\left(\frac{C|V|}{\delta}\right)}{\ln\left(\frac{d_{\max}}{d_{\max} - 1}\right)} \tag{8}$$

*the union of $T_1, T_2, \ldots, T_m$ contains every edge in $E$ with probability at least $1 - \delta$.*

*Proof.* By Lemma A.12 the probability that any edge $e$ appears in any DFS is at least $p_e \geq \frac{1}{d_{\max}}$. Hence, the probability that a single DFS tree does *not* contain $e$ is at most $1 - p_e \leq 1 - \frac{1}{d_{\max}}$. Since the spanning trees $T_1, T_2, \ldots, T_m$ are sampled independently, the probability that $e$ is missing from all $m$ trees is at most $\left(1 - \frac{1}{d_{\max}}\right)^m$. By the union bound over all $|E|$ edges, the probability that there exists at least one edge which is not covered by the union of the $m$ trees is at most

$$|E|\left(1 - \frac{1}{d_{\max}}\right)^m \leq C|V|\left(1 - \frac{1}{d_{\max}}\right)^m.$$

We require this probability to be at most $\delta$:

$$C|V|\left(1 - \frac{1}{d_{\max}}\right)^m \leq \delta.$$

Taking the natural logarithm on both sides gives:

$$\ln(C|V|) + m\ln\left(1 - \frac{1}{d_{\max}}\right) \leq \ln(\delta).$$

Since $\ln\left(1 - \frac{1}{d_{\max}}\right) < 0$, dividing by this term (and reversing the inequality) yields

$$m \geq \frac{\ln\left(\frac{C|V|}{\delta}\right)}{\ln\left(\frac{1}{1-\frac{1}{d_{\max}}}\right)} = \frac{\ln\left(\frac{C|V|}{\delta}\right)}{\ln\left(\frac{d_{\max}}{d_{\max}-1}\right)}.$$

Thus, with $m$ chosen accordingly, the union of the $m$ spanning trees contains every edge of $G$ with probability at least $1 - \delta$. $\qquad\square$

**Definition A.14** (Anonymous identity and parent tags). Let $G = (V, E, X)$ be a graph and let $S^{(1)}$ be the first search sampled on $G$ according to the RSNN search algorithm. Write $S^{(1)} = (w_0, w_1, \ldots, w_T)$ and let $\left(v_{(1)}, v_{(2)}, \ldots, v_{(n)}\right)$ be the vertices of $G$ ordered by their *first-visit time* along $S^{(1)}$, i.e. $v_{(i)}$ is the $i$-th distinct vertex encountered in $S^{(1)}$. Define the *anonymous identity tag* assignment $\tau : V \to [n]$ by

$$\tau\left(v_{(i)}\right) := i \qquad \text{for } i = 1, \ldots, n.$$

We use the same tag assignment $\tau$ for all searches in the RSNN search set on $G$. Because $S^{(1)}$ is sampled in a manner equivariant to vertex relabellings, the induced random tag assignment $\tau$ is permutation-invariant *in distribution*. Now let $S^{(j)}$ be any search produced by the RSNN search algorithm, and write $S^{(j)} = \left(u_0^{(j)}, u_1^{(j)}, \ldots, u_{T_j}^{(j)}\right)$. We define a *parent-tag sequence*

$$\pi^{(j)} = \left(\pi_0^{(j)}, \pi_1^{(j)}, \ldots, \pi_{T_j}^{(j)}\right) \in [n]^{T_j+1}$$

as follows. For each index $t$, let $q_t^{(j)}$ denote the vertex from which $u_t^{(j)}$ is discovered by the RSNN search algorithm along a true edge of $G$; when $u_t^{(j)}$ is chosen as an initial vertex of a search, we set $q_t^{(j)} := u_t^{(j)}$. We then define

$$\pi_t^{(j)} := \tau\left(q_t^{(j)}\right) \qquad \text{for } t = 0, \ldots, T_j.$$

In particular, the discovery vertex $q_t^{(j)}$ need not equal the previous vertex $u_{t-1}^{(j)}$ in the recorded sequence: $S^{(j)}$ may contain discontinuities where consecutive vertices $u_{t-1}^{(j)}$ and $u_t^{(j)}$ are not adjacent in $G$. The parent tag $\pi_t^{(j)}$ always records, in anonymous coordinates given by $\tau$, the vertex from which the search *actually* arrived at $u_t^{(j)}$ along a true edge of $G$. Thus, the collection of identity and parent tags $\left(\tau, (\pi^{(j)})_j\right)$ encodes all edges traversed by the RSNN searches, even across such discontinuities.

**Theorem A.15** (Universal Approximation by RSNNs on Sparse Graphs with Bounded Degree). *Let $\epsilon > 0$ and let $f : \mathcal{G} \to \mathbb{R}^d$ be any continuous graph-level function, where $\mathcal{G}$ is the space of sparse, unweighted graphs with $|E| = O(|V|)$, maximum degree at most $d_{\max}$, and size at most $|V| \leq n_{\max}$. Assume $f_{\mathrm{RSNN}}(G)$ uses (i) a universal set encoder $f_{\mathrm{agg}}$, (ii) a universal sequence encoder $f_{\mathrm{seq}}$, and (iii) anonymous identity and parent tags. Assume $m$ satisfies Lemma A.13, so that full coverage is achieved with probability at least $1 - \delta$. Then, with probability at least $1 - \delta$ there exists an RSNN configuration such that*

$$\|f_{\mathrm{RSNN}}(G) - f(G)\| < \epsilon \quad \text{for all } G \in \mathcal{G}, \tag{9}$$

*Proof.* Let $\mathcal{S}^{\mathrm{FC}}(G)$ be the set of search sets of size $m$ on $G$. Define a target on search sets by

$$\widetilde{f}(\mathcal{S}) := \begin{cases} F(G), & \mathcal{S} \in \mathcal{S}^{\mathrm{FC}}(G), \\ 0, & \text{otherwise.} \end{cases}$$

This $\widetilde{f}$ is well-defined (for any given input search set, there is a single unique output), and is permutation-invariant in the multiset argument. Because the input space (bounded-length sequences over a finite alphabet, aggregated into multisets of bounded size) is finite, the assumed universal sequence encoder and universal set aggregator can uniformly approximate $\widetilde{f}$ to error $< \varepsilon$ across $\bigcup_{G \in \mathcal{G}_{\leq n_{\max}}} \mathcal{S}^{\mathrm{FC}}(G)$. Therefore, with those parameters, for any $G$ and any random $\mathcal{S}(G)$,

$$\Pr\Big( \big\|f_{\mathrm{RSNN}}(\mathcal{S}) - F(G)\big\| < \varepsilon \ \Big| \ \mathcal{S} \in \mathsf{FullCov}(G) \Big) = 1,$$

and hence unconditionally $\Pr(\|f_{\mathrm{RSNN}}(\mathcal{S}) - F(G)\| < \varepsilon) \geq 1 - \delta$. $\qquad\square$

## A.3 Random Search Neural Network Invariance (Theorem 4.3, Corollary 4.4)

We next study invariance properties of RSNNs. Because RSNNs are randomized graph functions, we adopt a *probabilistic* notion of isomorphism invariance: if two graphs are isomorphic, the distributions of RSNN outputs coincide. As a consequence, the *expected* predictor $\Phi(G) = \mathbb{E}[f_{\mathrm{RSNN}}(G)]$ is an isomorphism-invariant graph function. Moreover, RSNNs *learn* this invariance via stochastic training: sampling a fresh search per step yields an unbiased gradient of the invariant risk, and under standard SGD conditions the parameters converge to a stationary point of the invariant objective. In practice, this justifies sampling with a small number of searches (e.g., $m{=}1$) in limited budget regimes.

**Theorem A.16** (Isomorphism-Invariance of RSNN). *A randomized search procedure on a graph $G$ produces a sequence $S^G = (s_0^G, \dots, s_{|V(G)|}^G)$ of visited vertices. We say the procedure is probabilistically invariant to graph isomorphisms if,*

$$\big(\pi(s_0^G), \dots, \pi(s_{|V(G)|}^G)\big) \ \overset{d}{=} \ (s_0^H, \dots, s_{|V(H)|}^H) \quad \text{for all } G \overset{\pi}{\cong} H.$$

*The randomized DFS procedure used in RSNNs satisfies the above definition. Hence, RSNNs satisfy probabilistic invariance: for all $G \cong H$, $f_{\mathrm{RSNN}}(G) \overset{d}{=} f_{\mathrm{RSNN}}(H)$, and the averaged predictor $\Phi(G) := \mathbb{E}\big[f_{\mathrm{RSNN}}(G)\big]$ is an invariant function on graphs: $\Phi(G) = \Phi(H)$ for all $G \cong H$.*

*Proof.* Write $X_{\mathrm{DFS}}(G) = (s_0, \dots, s_{|V|-1})$ for the vertex sequence produced by the randomized DFS on $G$, and let $H = \pi \cdot G$ for an isomorphism $\pi : V(G) \to V(H)$. The randomness comes from: (i) the root $s_0 \sim \mathrm{Unif}(V(G))$ and (ii) an independent random order of neighbors at each vertex.

We prove by induction on $t$ that the next state has the same *pushforward* conditional law under any isomorphism $\pi$:

$$\pi\big(X_{\mathrm{DFS}}(G)[t] \,\big|\, \mathbf{x}\big) \ \overset{d}{=} \ X_{\mathrm{DFS}}(H)[t] \,\big|\, \pi\mathbf{x}, \tag{10}$$

for every valid DFS prefix $\mathbf{x} = (s_0, \dots, s_{t-1})$ on $G$ (and its image $\pi\mathbf{x}$ on $H$). Averaging over prefixes then yields $\pi X_{\mathrm{DFS}}(G)[t] \overset{d}{=} X_{\mathrm{DFS}}(H)[t]$ for each $t$, and thus $\pi X_{\mathrm{DFS}}(G) \overset{d}{=} X_{\mathrm{DFS}}(H)$.

**State, admissible set, and frontier.** For a prefix $\mathbf{x}$ valid on $G$, let $V_{\text{vis}}(G; \mathbf{x}) = \{s_0, \ldots, s_{t-1}\}$ be the visited set and let $\text{top}(G; \mathbf{x})$ be the current DFS stack top (the vertex whose adjacency list is being explored). Define the *admissible neighbor set*

$$A(G; \mathbf{x}) := \mathcal{N}\big(\text{top}(G; \mathbf{x})\big) \setminus V_{\text{vis}}(G; \mathbf{x}).$$

If $A(G; \mathbf{x}) \neq \varnothing$, the rule "pick the unvisited neighbor at random" makes the next vertex $s_t$ *uniform* on $A(G; \mathbf{x})$. If $A(G; \mathbf{x}) = \varnothing$, the next move is the (deterministic) backtrack to the parent of $\text{top}(G; \mathbf{x})$ in the current DFS tree. Under an isomorphism $\pi : G \cong H$, relabeling preserves these invariants:

$$\text{top}(H; \pi\mathbf{x}) = \pi\big(\text{top}(G; \mathbf{x})\big), \quad V_{\text{vis}}(H; \pi\mathbf{x}) = \pi\big(V_{\text{vis}}(G; \mathbf{x})\big), \quad A(H; \pi\mathbf{x}) = \pi\big(A(G; \mathbf{x})\big).$$

**Base case ($t = 0$).** $s_0 \sim \text{Unif}(V(G))$ and $\pi s_0 \sim \text{Unif}(V(H))$, so

$$\pi X_{\text{DFS}}(G)[0] \stackrel{d}{=} X_{\text{DFS}}(H)[0].$$

**Induction step.** Assume $\pi X_{\text{DFS}}(G)[: t] \stackrel{d}{=} X_{\text{DFS}}(H)[: t]$. Fix any realization $\mathbf{x}$ of the prefix on $G$. There are two cases.

*(i) Expansion step:* $A(G; \mathbf{x}) \neq \varnothing$. Conditioned on $\mathbf{x}$, $X_{\text{DFS}}(G)[t]$ is uniform on $A(G; \mathbf{x})$. Conditioned on $\pi\mathbf{x}$, $X_{\text{DFS}}(H)[t]$ is uniform on $A(H; \pi\mathbf{x}) = \pi A(G; \mathbf{x})$. Pushing the uniform measure on $A(G; \mathbf{x})$ forward by $\pi$ yields the uniform measure on $\pi A(G; \mathbf{x})$, hence

$$\pi\big(X_{\text{DFS}}(G)[t] \mid \mathbf{x}\big) \stackrel{d}{=} X_{\text{DFS}}(H)[t] \mid \pi\mathbf{x}.$$

*(ii) Backtrack step:* $A(G; \mathbf{x}) = \varnothing$. The next state is the parent of $\text{top}(G; \mathbf{x})$ in the DFS tree determined by $\mathbf{x}$; thus it is *deterministic* given $\mathbf{x}$. Relabeling preserves parent/child relations in the explored DFS tree, so

$$\pi\big(X_{\text{DFS}}(G)[t] \mid \mathbf{x}\big) = X_{\text{DFS}}(H)[t] \mid \pi\mathbf{x},$$

In both cases, the conditional laws match after applying $\pi$. Taking expectations over the distributions gives $\pi X_{\text{DFS}}(G)[t] \stackrel{d}{=} X_{\text{DFS}}(H)[t]$ for each $t$, which completes the induction and yields

$$\pi X_{\text{DFS}}(G) \stackrel{d}{=} X_{\text{DFS}}(H).$$

This proves probabilistic invariance of the randomized DFS. Since the RSNN output $f_{\text{RSNN}}$ is a deterministic function of the search sequence, it follows that $f_{\text{RSNN}}(G) \stackrel{d}{=} f_{\text{RSNN}}(H)$, and the averaged predictor $\Phi(G) = \mathbb{E}[f_{\text{RSNN}}(G)]$ is an invariant graph function. $\square$

**Corollary A.17** (Stochastic training converges to the invariant objective). *Let $\ell(\cdot, y)$ be a differentiable loss. Consider the expected risk*

$$L(\mathbf{W}) = \mathbb{E}_{(G,y) \sim \mathcal{D}} \, \mathbb{E}_{S \sim \mathcal{S}_{\text{DFS}}(G)} \big[\ell\big(f_{\text{RSNN}}(G, S; \mathbf{W}), y\big)\big].$$

*At each SGD step $t$, sample $(G_t, y_t) \sim \mathcal{D}$ and one search draw $S_t \sim \mathcal{S}_{\text{DFS}}(G_t)$, and update*

$$\mathbf{W}_{t+1} = \mathbf{W}_t - \eta_t \, \nabla_{\mathbf{W}} \, \ell\big(f_{\text{RSNN}}(G_t, S_t; \mathbf{W}_t), y_t\big).$$

*Then $\mathbb{E}\big[\nabla_{\mathbf{W}} \ell(f_{\text{RSNN}}(G_t, S_t; \mathbf{W}_t), y_t)\big] = \nabla_{\mathbf{W}} L(\mathbf{W}_t)$, i.e., the single-sample gradient is an unbiased estimator of the invariant objective's gradient. Under standard SGD conditions, $\mathbf{W}_t$ converges almost surely to the optimal $\mathbf{W}^\star$ of the invariant objective.*

*Proof sketch.* This follows directly from the proof of Proposition A.1 in Murphy et al. [29], replacing permutations by RSNN searches: since the search randomness $S \sim \mathcal{S}_{\text{DFS}}(G)$ is sampled independently of $\mathbf{W}$ and $\ell$ is differentiable with integrable gradient, we can exchange $\nabla_{\mathbf{W}}$ and the expectations to get $\nabla_{\mathbf{W}} L(\mathbf{W}) = \mathbb{E}_{(G,y) \sim \mathcal{D}} \mathbb{E}_{S \sim \mathcal{S}_{\text{DFS}}(G)} \big[\nabla_{\mathbf{W}} \ell(f_{\text{RSNN}}(G, S; \mathbf{W}), y)\big]$, so the single-sample stochastic gradient is unbiased; standard Robbins–Monro/Polyak supermartingale arguments then yield a.s. convergence of SGD to a stationary point (and to $\mathbf{W}^\star$ under convexity). $\square$

# B Additional Model Details: Positional Encodings and Sampling Algorithms

In this section, we provide additional details on the positional encoding scheme and sampling algorithms used in both RSNN and RWNN models. These components are essential not only for implementation but also for theoretical expressivity. We also present detailed descriptions of the sampling procedures for both random walks and random searches. For RWNNs, we outline the walk generation algorithm, including initialization, neighbor selection, and PE encoding. For RSNNs, we describe the random depth-first search strategy, including how spanning trees are constructed and how node visitation is handled.

## B.1 Positional Encodings

**Identity and Adjacency Encodings.** Tönshoff et al. [5] and Chen et al. [6] augment each walk with two binary feature matrices that inject explicit structural context. For a walk $W = (w_0, \ldots, w_\ell)$ on graph $G$, the *identity encoding* $\mathrm{id}_W^s \in \{0,1\}^{(\ell+1)\times s}$ marks node repetitions within a sliding window of size $s$: for indices $0 \leq i \leq \ell$ and $0 \leq j \leq s-1$ we set

$$\mathrm{id}_W^s[i,j] = 1 \text{ iff } i - j \geq 1 \text{ and } w_i = w_{i-j},$$

and 0 otherwise. Thus column $j$ signals whether the current node re-appeared exactly $j$ steps earlier, explicitly encoding cycles of length $j+1$. Second, the *adjacency encoding* $\mathrm{adj}_W^s \in \{0,1\}^{(\ell+1)\times(s-1)}$ records edges among already-visited nodes that the walk does not traverse. We define

$$\mathrm{adj}_W^s[i,j] = 1 \text{ iff } i - j \geq 1 \text{ and } (w_i, w_{i-j}) \in E(G),$$

and 0 otherwise. Here, $E(\cdot)$ denotes the edge set of the input. Consequently, for every pair of nodes that appears within the window, the encoding reveals whether they are adjacent in the underlying graph. The two blocks are concatenated to form a positional-encoding matrix $h_{\mathrm{PE}} = [\,\mathrm{id}_W^s \,\|\, \mathrm{adj}_W^s\,] \in \mathbb{R}^{(\ell+1)\times d_{\mathrm{pe}}}$ with $d_{\mathrm{pe}} = 2s - 1$. Appending $h_{\mathrm{PE}}$ to the raw node embeddings ensures that, once full node- and edge-coverage is achieved, the sequence model receives enough information to reconstruct the entire subgraph induced by the walk.

**Anonymous Encodings.** As an alternative to the identity–adjacency scheme, *anonymous encodings* have been proposed to capture graph structure by Wang and Cho [3] and Kim et al. [7]. For a walk $W$ we create an integer vector $\omega_{\mathrm{anon}}(W) \in \{1, \ldots, \ell+1\}^{\ell+1}$ defined recursively:

$$\omega_{\mathrm{anon}}(W)[t] = \begin{cases} 1 + \max\{\,\omega_{\mathrm{anon}}(W)[0{:}t-1]\,\}, & \text{if } w_t \notin \{w_0, \ldots, w_{t-1}\}, \\ \omega_{\mathrm{anon}}(W)[s], & \text{if } s < t \text{ is the first index with } w_s = w_t. \end{cases}$$

In words, the first time a node appears in the walk it is assigned the next unused label $1, 2, 3, \ldots$; every subsequent visit to that same node reuses the original label. Hence $\omega_{\mathrm{anon}}$ is invariant to the specific node IDs yet records the order in which *unique* vertices are discovered, providing topological context without relying on absolute labels.

**Role in Expressivity.** These positional encodings play a critical role in the expressive power of both RWNNs and RSNNs. They serve as the main mechanism by which the walk or search encodes structural information from the underlying graph. In particular, the identity and anonymous encodings, when combined with walks that achieve full edge coverage, allow for exact reconstruction of the input graph. Meanwhile, adjacency encodings with sufficient window size and identity/parent tags introduced in Definition A.14 enable full reconstruction even with only node coverage, as they record structural edges not explicitly traversed in the sequence. In our RSNN implementation, we rely on adjacency encodings. These are especially important for preserving expressivity in RSNNs: depth-first searches introduce disconnections in the sequence, where jumps between non-adjacent nodes may obscure structure. Consider for example nodes $w_i$ and $w_{i+1}$ traversed adjacent to one another in a search sequence, but disconnected in the graph. With an appropriate window size, the adjacency encoding first signals the disconnection setting $\mathrm{adj}_W^s[i+1,1] = 0$, then identifies the connecting edge when it appeared in the sequence setting $\mathrm{adj}_W^s[i+1,j] = 1$ for $(w_i, w_{i-j}) \in E(G)$. This ensures that, once full edge coverage is achieved across searches, the sequence model receives all structural information necessary to reconstruct the graph. Thus, positional encodings are central to the theoretical guarantees of RSNN expressivity.

---

**Algorithm 1:** Uniform Random Walk with Positional Encodings

---

**Input:** Graph $G = (V, E)$, walk length $l$, window size $s$
**Output:** Random walk $W = (w_0, \ldots, w_l)$, encodings $\mathrm{id}_W^s$, $\mathrm{adj}_W^s$
Sample initial node $w_0 \sim \mathcal{U}(V)$
Initialize $W \leftarrow [w_0]$
**for** $i \leftarrow 1$ *to* $l$ **do**
    Let $\mathcal{N}(w_{i-1})$ be the neighbors of $w_{i-1}$
    Sample $w_i \sim \mathcal{U}(\mathcal{N}(w_{i-1}))$
    Append $w_i$ to $W$
    **for** $j \leftarrow 1$ *to* $s$ **do**
        **if** $i - j \geq 0$ **then**
            $\mathrm{id}_W^s[i, j] \leftarrow \mathbf{1}[w_i = w_{i-j}]$                      `// Identity encoding`
            $\mathrm{adj}_W^s[i, j] \leftarrow \mathbf{1}[(w_i, w_{i-j}) \in E]$             `// Adjacency encoding`

**return** $W$, $\mathrm{id}_W^s$, $\mathrm{adj}_W^s$

---

---

**Algorithm 2:** Random Depth-First Search with Adjacency Encodings

---

**Input:** Graph $G = (V, E)$, window size $s$
**Output:** Search sequence $W = (w_0, \ldots, w_\ell)$, adjacency encoding $\mathrm{adj}_W^s$
Sample initial node $w_0 \sim \mathcal{U}(V)$
Initialize stack $\mathcal{S} \leftarrow [w_0]$, visited set $\mathcal{V} \leftarrow \{w_0\}$, walk $W \leftarrow [\,]$
Initialize $\mathrm{adj}_W^s \leftarrow \mathbf{0}^{|V| \times (s-1)}$
**while** $\mathcal{S}$ *is not empty* **do**
    Pop $u \leftarrow \mathcal{S}$
    Append $u$ to $W$
    **for** $j \leftarrow 1$ **to** $s - 1$ **do**
        **if** $|W| > j$ **then**
            $\mathrm{adj}_W^s[|W| - 1, j] \leftarrow \mathbf{1}[(u, W[|W| - 1 - j]) \in E]$       `// Adjacency encoding`
    Let $\mathcal{N}(u)$ be unvisited neighbors of $u$ in random order
    **foreach** $v \in \mathcal{N}(u)$ **do**
        Push $v$ onto $\mathcal{S}$
        Add $v$ to $\mathcal{V}$

**return** $W$, $\mathrm{adj}_W^s$

---

### B.2 Sampling Algorithms

**Random Walk Sampling.** We adopt a standard uniform random walk procedure to extract sequences from a graph (Algorithm 1). The algorithm begins by uniformly sampling a starting node from the vertex set. At each step, it selects the next node uniformly at random from the current node's neighbors. As the walk progresses, we maintain a sliding window of fixed size $s$ to compute identity and adjacency encodings for each step. The algorithm takes as input the graph $G$, walk length $l$, and window size $s$, and returns both the walk and the corresponding structural encodings.

**Random Search Sampling.** We implement random searches in RSNNs using a randomized depth-first search (DFS) traversal (Algorithm 2). The algorithm begins by sampling a starting node uniformly at random from the vertex set. From there, we perform a standard DFS, visiting each neighbor in a random order to introduce stochasticity. As nodes are visited, they are recorded sequentially in the walk $W$, and only the adjacency-based positional encoding $\mathrm{adj}_W^s$ is computed using a sliding window of size $s$. Since DFS visits each node exactly once, identity encodings are unnecessary. The resulting walk and adjacency encoding together define the structural input for RSNNs.

# C  Extended Results

We present two additional experiments to complement our main findings. First, we conduct an ablation study evaluating the impact of the sequence model architecture on performance by comparing CRAWL, the best performing RWNN, and RSNNs equipped with GRUs, LSTMs, and Transformers on molecular benchmarks. This experiment helps assess whether the RSNN framework is sensitive to the choice of sequence model. Second, we report runtime comparisons between RSNNs and RWNNs to evaluate computational efficiency. Specifically, we compare training times across varying sample sizes to understand how the two approaches scale under realistic computational budgets.

## C.1  Sequence Model Ablations

We evaluate the impact of sequence model architecture by comparing RSNNs and CRAWL equipped with GRUs, LSTMs, and Transformers (Table 1). Across all configurations, the trends from the main paper hold: RSNNs consistently outperform RWNNs at low sample sizes ($m = 1$), regardless of sequence model. Notably, RSNNs with $m = 1$ often match or exceed the performance of RWNNs with $m = 16$, reaffirming the sample efficiency advantages of random search. When $m = 16$ on the **BACE** dataset, CRAWL-LSTM and CRAWL-GRU slightly outperform their RSNN counterparts, however in the remaining comparisons RSNN always outperforms CRAWL across all $m$. Overall, GRUs and LSTMs perform comparably within both RSNN and RWNN variants, indicating that RSNN improvements are robust to the choice of sequence model, provided it has adequate recurrence-based inductive bias. In contrast, Transformers underperform relative to GRUs and LSTMs across most benchmarks and sample sizes. One possible explanation is that Transformers lack the hard-coded recurrence structure present in GRUs and LSTMs, relying instead on global attention mechanisms

Table 4: Median (min, max) of model AUC across test splits on molecular benchmarks. We report results for each model equipped with one of three sequence models: (1) GRU, (2) LSTM, or (3) Transformer (TRSF), as indicated by the suffix. The best model for each value of $m$ is highlighted in **blue**. Trends from the main paper hold across architectures: RSNNs consistently outperform RWNNs at low sample sizes, with GRUs and LSTMs yielding similar performance, while Transformers underperform across most settings.

| | | MoleculeNet Molecular Benchmarks (AUC ↑) | | | | | |
|---|---|---|---|---|---|---|---|
| | | **CLINTOX** | **SIDER** | **BACE** | **BBBP** | **TOX21** | **TOXCAST** |
| | **# Graphs** | 1.5K | 1.5K | 1.5K | 2K | 8K | 9K |
| | **Avg. $|V|$** | 26.1 | 33.6 | 34.1 | 23.9 | 18.6 | 18.8 |
| | **Avg. $|E|$** | 55.5 | 70.7 | 73.7 | 51.6 | 38.6 | 38.5 |
| | **# Classes** | 2 | 2 | 2 | 2 | 2 | 2 |
| $m = 1$ | **CRAWL-TRSF** | 59.8 (48.1, 71.8) | 60.3 (57.2, 68.4) | 67.6 (65.2, 73.3) | 74.6 (66.1, 79.4) | 70.4 (65.3, 74.5) | 70.8 (65.4, 75.3) |
| | **CRAWL-LSTM** | 66.7 (40.4, 80.2) | 61.4 (57.4, 63.8) | 66.2 (60.7, 71.4) | 74.4 (68.4, 80.4) | 72.2 (67.6, 76.0) | 71.5 (67.7, 75.4) |
| | **CRAWL-GRU** | 70.0 (64.6, 73.6) | 64.2 (56.1, 67.2) | 62.5 (59.2, 70.8) | 77.6 (68.8, 81.5) | 71.7 (66.4, 75.3) | 72.8 (67.7, 76.7) |
| | **RSNN-TRSF** | 82.9 (59.8, 87.9) | 65.6 (63.1, 71.9) | 78.0 (71.3, 81.5) | 85.6 (77.6, 89.8) | 77.7 (73.8, 78.9) | 74.2 (70.8, 78.8) |
| | **RSNN-LSTM** | 87.2 (82.6, 89.4) | **66.8 (61.7, 72.2)** | 78.2 (74.3, 84.3) | 87.1 (83.9, 89.5) | 79.5 (77.2, 83.7) | **75.6 (72.9, 80.6)** |
| | **RSNN-GRU** | **88.1 (84.9, 91.5)** | 66.2 (63.0, 72.4) | **79.7 (75.9, 83.6)** | **87.5 (80.3, 89.9)** | **79.8 (77.2, 83.4)** | 74.6 (72.3, 79.7) |
| $m = 4$ | **CRAWL-TRSF** | 69.4 (49.0, 79.0) | 64.7 (61.1, 69.5) | 73.7 (68.4, 75.4) | 82.6 (77.5, 87.7) | 74.5 (71.6, 78.6) | 71.3 (69.1, 80.0) |
| | **CRAWL-LSTM** | 80.4 (72.3, 83.8) | 66.3 (63.2, 68.8) | 72.7 (67.5, 78.5) | 84.0 (78.5, 88.6) | 77.5 (75.3, 79.9) | 74.6 (71.1, 79.9) |
| | **CRAWL-GRU** | 83.0 (76.6, 91.5) | 65.2 (59.5, 71.3) | 75.7 (71.0, 79.0) | 84.5 (80.7, 87.0) | 77.6 (75.6, 81.2) | 74.4 (69.2, 77.9) |
| | **RSNN-TRSF** | 84.2 (63.4, 87.0) | 67.1 (64.6, 70.8) | 79.8 (69.4, 82.5) | 85.6 (79.9, 90.7) | 78.0 (74.2, 83.0) | 76.6 (71.5, 81.2) |
| | **RSNN-LSTM** | 88.7 (81.2, 90.8) | **67.5 (64.1, 70.1)** | **80.9 (75.3, 84.4)** | **88.9 (82.7, 91.6)** | **81.4 (76.3, 83.3)** | **76.6 (73.8, 81.3)** |
| | **RSNN-GRU** | **89.1 (80.9, 91.7)** | 67.0 (61.3, 71.1) | 80.4 (76.5, 84.0) | 88.0 (80.3, 90.5) | 80.3 (77.3, 84.2) | 76.1 (72.2, 79.0) |
| $m = 8$ | **CRAWL-TRSF** | 68.3 (53.1, 88.1) | 65.9 (62.6, 71.4) | 75.4 (66.6, 80.7) | 85.4 (79.2, 89.6) | 76.4 (71.8, 78.2) | 75.2 (72.0, 78.7) |
| | **CRAWL-LSTM** | 87.2 (78.3, 89.4) | 67.1 (63.6, 70.7) | 79.2 (76.8, 83.2) | 86.8 (79.5, 91.6) | 78.9 (76.0, 81.7) | 73.5 (68.9, 77.3) |
| | **CRAWL-GRU** | 86.5 (83.6, 91.4) | 66.1 (62.1, 69.6) | 80.3 (71.0, 82.5) | 86.0 (82.8, 89.6) | 79.1 (76.7, 82.1) | 75.5 (72.0, 78.6) |
| | **RSNN-TRSF** | 82.7 (51.8, 89.9) | 66.8 (62.5, 72.0) | 80.2 (73.3, 82.4) | 86.4 (79.8, 90.7) | 76.8 (75.4, 81.5) | 75.2 (71.5, 81.4) |
| | **RSNN-LSTM** | **88.4 (82.2, 90.6)** | 67.2 (64.3, 74.6) | **80.7 (74.8, 87.1)** | 88.1 (82.6, 91.4) | 81.1 (77.7, 85.2) | **75.9 (72.3, 82.2)** |
| | **RSNN-GRU** | 88.3 (80.1, 91.3) | **67.6 (63.3, 69.2)** | 80.0 (76.1, 85.1) | **88.6 (83.6, 90.3)** | **82.2 (77.3, 85.3)** | 75.7 (73.0, 78.9) |
| $m = 16$ | **CRAWL-TRSF** | 69.6 (47.6, 86.9) | 65.1 (63.1, 70.1) | 78.8 (73.5, 79.7) | 85.2 (79.5, 89.3) | 77.7 (75.8, 81.9) | 74.8 (72.1, 80.0) |
| | **CRAWL-LSTM** | 87.8 (80.1, 89.5) | 65.7 (63.4, 69.0) | 79.5 (74.4, 86.0) | 87.1 (79.7, 93.9) | 79.2 (77.9, 82.3) | 76.2 (70.4, 79.0) |
| | **CRAWL-GRU** | 89.1 (80.5, 91.1) | 65.3 (61.4, 70.8) | 80.7 (76.1, 84.5) | 87.0 (81.7, 90.3) | 80.9 (77.4, 82.6) | 76.2 (72.7, 77.9) |
| | **RSNN-TRSF** | 84.4 (78.5, 91.7) | 66.6 (63.6, 73.6) | 81.0 (73.1, 82.8) | 86.0 (78.7, 90.7) | 77.6 (74.5, 82.1) | 76.4 (72.3, 79.2) |
| | **RSNN-LSTM** | 88.3 (81.9, 92.2) | **67.3 (64.8, 71.9)** | **80.5 (79.0, 84.3)** | 88.5 (83.8, 91.2) | 82.0 (78.8, 83.5) | 75.5 (72.9, 80.0) |
| | **RSNN-GRU** | **88.5 (82.0, 93.7)** | 67.1 (65.0, 74.0) | 79.8 (76.8, 84.9) | **89.4 (83.0, 91.7)** | **82.2 (78.0, 84.1)** | **76.5 (73.4, 79.3)** |

that may require more data to model sequential dependencies effectively, especially in low-sample regimes. These results suggest that recurrent sequence models are better suited for graph-based walk or search processing under constrained sampling budgets.

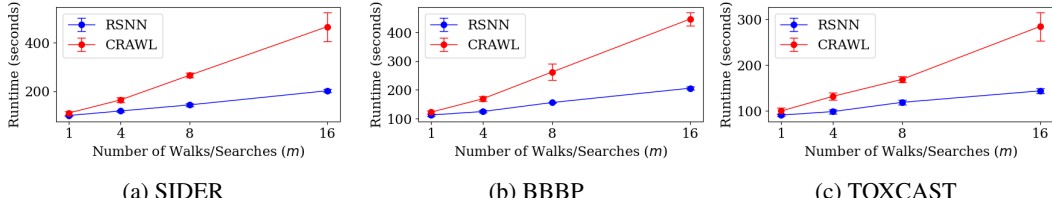

|  (a) SIDER | (b) BBBP | (c) TOXCAST |

Figure 4: Training runtime (in seconds) of RSNN and CRAWL over 25 epochs on **SIDER**, **BBBP**, **TOXCAST** as a function of the number of samples $m$. Error bars represent standard deviation across 5 runs. At low sample counts, RSNNs exhibit comparable runtime to RWNNs; as $m$ increases, RSNNs become faster despite longer sequence lengths. We hypothesize this is due to random walks repeatedly visiting high-degree nodes, incurring more computation per step, whereas DFS-based searches visit each node exactly once.

## C.2 Runtime Comparisons

**Experimental Setup.** To ensure a fair comparison between RSNNs and CRAWL, we fix all model components except for the sampling strategy. Both models use a GRU sequence model with hidden dimension 64 and are composed of 2 layers. We set the positional encoding window size to 8 and batch size to 64. For each graph, the random walk length is set to $l = |V|$, equal to the number of nodes, so that the number of sequence steps is identical between random walks and searches. As a result, RSNNs and CRAWL have equivalent asymptotic runtimes per sample. We measure training runtime over 25 epochs on three molecular benchmarks, **SIDER**, **BBBP**, and **TOXCAST**, across varying sample sizes $m \in \{1, 4, 8, 16\}$. For each forward pass, a fresh set of $m$ walks or searches is sampled per graph. All experiments are run on a single NVIDIA GeForce GTX 1080 Ti GPU, with sampling parallelized across 4 CPU workers to reflect practical deployment conditions.

**Results.** Empirically we observe that RSNN searches are never more expensive than CRAWL walks for any tested number of walks $m$, and that for larger $m$ the RSNN implementation can even become faster (Figure 4). Although, each routine shares the same asymptotic cost, $\mathcal{O}(|V|)$ on our sparse graphs, they differ by constant factors that affect runtime comparisons in practice:

- **Random Walks Revisit Nodes with Larger Degrees.** A DFS visits each vertex exactly once, while a random walk visits nodes randomly, potentially revisiting many vertices with higher degrees. Consequently, searches and random walks visit different sets of nodes. This affects runtime since operations per node depend on their degrees (e.g., shuffling neighbors, random choices on neighbors, identity/adjacency checks), incurring more computation per-step and increasing runtimes for random walks.

- **Per–step work.** The DFS runs one `for s` loop that updates a *single* adjacency-encoding tensor. The RWNN walk performs an identical `for s` loop, but each iteration evaluates *two* conditions (identity & adjacency encoding) and writes to *two* tensors, effectively doubling the cost of that inner loop per step.

- **Neighbor handling.** DFS shuffles the neighbor list once per new vertex, whereas random walks rebuilds a Python list and calls `random.choice` at every step, and if non-backtracking is enabled, creates an additional filtered list. These repeated list allocations and Python-level random picks inflate wall time.

Together, these constant-factor differences explain why the asymptotically identical $\mathcal{O}(|V|)$ algorithms show distinct wall-time profiles: RSNN remains competitive for all $m$, while CRAWL exhibit longer runtimes at larger $m$.

# D  Experimental Details and Code

**Training and Hyperparameter Selection.**    All models are trained by minimizing the binary cross-entropy loss on molecular benchmarks and the negative log-likelihood loss on protein benchmarks. Training is performed for a maximum of 200 epochs with early stopping patience set to 25 epochs based on validation performance. The best-performing model on the validation set is selected for evaluation on the test set. We perform a grid search over the following hyperparameters for all RWNN and RSNN models:

- Number of layers: $\{1, 2, 3\}$
- Learning rate: $\{0.05, 0.01, 0.005, 0.001\}$
- Batch size: $\{32, 64, 128\}$
- Hidden dimension: $\{32, 64, 128\}$
- Global pooling: $\{$`mean`, `sum`, `max`$\}$
- Sequence model: $\{$`GRU`, `LSTM`, `Transformer`$\}$
- Number of samples $m$: $\{1, 4, 8, 16\}$

We fix the window size $s = 8$ for both CRAWL and RSNN models. All models are optimized using the Adam optimizer [42].

# E  Extended Discussion

**Background on WL and its Variants.**    The Weisfeiler–Lehman (WL) hierarchy has become a standard lens for characterizing graph model expressivity. Xu et al. [15] first established the equivalence between 1-dimensional WL and MPNNs, while Morris et al. [39], Azizian and Lelarge [21] generalized this perspective to higher-order GNNs via higher-order WL variants. Beyond MPNNs, recent work has aligned graph transformers with WL, clarifying their expressivity within the same hierarchy [43, 44]. In parallel, random walk kernels and path GNNs have been connected to WL as sequence-based representations [41, 45].

Our *Walk Weisfeiler–Lehman* (WWL) refinement builds directly on this line: we introduce a walk-based color refinement and show that, under full coverage, its distinguishing power coincides with 1-WL. In doing so, we place RWNNs firmly within the WL-centered expressivity landscape alongside MPNNs, graph transformers, and path-based GNNs, advancing a unified view of diverse graph learning architectures through the WL hierarchy.

