# OpenReview forum: "Random Search Neural Networks for Efficient and Expressive Graph Learning"
_NeurIPS.cc/2025/Conference — NeurIPS 2025 poster_

### Official Review · Reviewer_jLHR · 2025-06-30

**Clarity:** 2
**Significance:** 3
**Originality:** 3
**Rating:** 4
**Confidence:** 2

**Summary:**

This paper introduces Random Search Neural Networks (RSNNs), a novel graph learning framework designed to overcome the expressive limitations of existing Random Walk Neural Networks (RWNNs). The authors argue that under realistic sampling constraints, RWNNs often fail to achieve complete node and edge coverage of a graph, thereby limiting their ability to capture global structure. To address this, RSNNs replace the random walk sampling strategy with a random search procedure, specifically depth-first search (DFS).

The core contributions of the paper are both theoretical and empirical:

1. Theoretical Analysis: The article theoretically proves that the RSNNs are more expressive and have tighter generalization.

2. Empirical Validation: Through experiments on molecular and protein graph classification benchmarks, RSNNs are shown to consistently and significantly outperform various RWNN baselines.

**Questions:**

1. For the experiments, the RWNN walk length $I$ was set to |V| to ensure equivalent asymptotic runtimes with DFS. However, RWNNs are often used with much shorter walks [1]. So, RWNN walk length set to |V| may increase time complexity, but brings a slight performance improvement. Could the authors provide a comparison against RWNNs in this more conventional, computationally cheaper setting? This would offer a more direct view of the trade-off between computational cost, coverage, and performance.

2. The article theoretically proves Tighter Learning Guarantees in RSNNs. Why not provide experimental evidence to further confirm this?

3. RWNN has poor expressiveness due to limited coverage, so you compare it with subgraph GNN in Section 3.1. I want to know why the subgraph GNN is not used as the baseline model in the experiment.

Reference:

[1] Nikolentzos, Giannis, and Michalis Vazirgiannis. "Random walk graph neural networks." Advances in Neural Information Processing Systems 33 (2020): 16211-16222.

**Ethical Concerns:**

["NO or VERY MINOR ethics concerns only"]

**Final Justification:**

This paper introduces RSNN, a model designed to improve the ability of GNN representation. The methodology is clearly described, and the experimental results are solid and demonstrate a clear improvement over existing baselines. In the rebuttal stage, the author efficiently addresses my problem.
While some details e.g. theorem statement, technical implementation, could be improved, these make it easier for readers to read.
Overall, I believe this paper makes a meaningful contribution and could be accepted

**Limitations:**

Yes

**Paper Formatting Concerns:**

There is no major formatting issue, except that the proofs in the appendix could be organized to make them easier to read.

**Quality:**

3

**Strengths And Weaknesses:**

Strengths:

Impressive Sample Efficiency: The empirical results demonstrate the practical value of RSNNs. The finding that an RSNN with a single search (m=1) can match or outperform the best RWNN with 16 walks (m=16) on several benchmarks is a standout result that underscores the method's efficiency.

Strong Theoretical Foundation: The proof that partially-covered RWNNs are less expressive than subgraph GNNs (Theorem 3.1) provides a motivation for the work. The subsequent theorems establishing the efficient coverage (Lemma 4.1), universal approximation capability (Theorem 4.2), and tighter generalization bounds (Theorem 4.3) of RSNNs provide a solid theoretical underpinning for the proposed method.

Weaknesses:

Scalability to Dense Graphs: The primary weakness is the reliance on a full depth-first search. While efficient on the sparse graphs targeted in the paper, the O(|V|+|E|) complexity of DFS would be prohibitive for very large or dense graphs. This limits the immediate applicability of the current RSNN framework to a subset of all possible graph structures.

---

> ### Author Rebuttal · Authors · 2025-07-30
>
> We appreciate your thoughtful and careful feedback. We are pleased that you found our theoretical foundation strong, providing a solid foundation for RSNNs, and our empirical results on the practical value of RSNNs impressive. Below, we address your specific comments individually.
>
> - **How does RSNN scale to large and dense graphs?** Great question. We ran additional experiments to measure how RSNN scales to **dense** graphs (see experiments below). We don’t evaluate on large graphs because we focus our evaluation on small to medium graphs, where global structure, long-range dependencies, and high expressivity are essential, in comparison to large graphs where locality is more important to capture and as a result message-passing GNNs already work well (e.g., node classification in citation networks). To demonstrate our approach is scalable to **dense** graphs, we have since evaluated RSNN and CRAWL, the strongest RWNN baseline, on two dense graph classification benchmarks from NeuroGraph (Said et al., 2023): a Gender prediction set of 1000 graphs (avg. 1000 nodes, avg. 46000 edges, avg. degree ≈ 46) and an Activity prediction set of 7500 graphs (avg. 400 nodes, avg. 7000 edges, avg. degree ≈ 18), both substantially denser than our molecular and protein datasets. Across all sample lengths $m$, RSNN consistently outperforms CRAWL in accuracy (Table 1 in response to Reviewer k2UB) while remaining comparable to CRAWL runtime. On the Activity dataset, sixteen RSNN searches complete in 0.24 seconds compared to 0.32 seconds for sixteen CRAWL walks; on the denser Gender dataset, sixteen searches take 1.10 seconds versus 1.03 seconds for sixteen walks. These results show that even with DFS cost RSNN achieves comparable runtimes to RWNNs in much denser graphs, underscoring its scalability and broad applicability across a spectrum of graph densities.
>
> - **Could the authors provide a comparison against RWNNs in a more conventional, computationally cheaper setting?** Yes, we conduct an additional experiment testing different walk lengths $\ell$ fixing $m=4$ for RWNN on the CLINTOX dataset, measuring performance, average node coverage, and runtime for 50 train epochs. We find that shorter walks indeed yield faster sampling times but also incur lower coverage and consequently lower performance, highlighting the balance between computational efficiency and expressivity. These additional results will be included in the revised manuscript to offer a comprehensive comparison of the tradeoffs between the three factors.
>
> Table 3. Runtime, coverage, and performance as $\ell$ increases on CLINTOX fixing $m=4$ for RWNN.
> | CLINTOX | $\ell$=4 | $\ell$=8 | $\ell$=16 | $\ell=26=\|V_{avg}\|$ |
> | ------- | ------- | ------- | ------- | ------- |
> |    AUC     |    0.731 ± 0.049     |     0.757 ± 0.065    |     0.798 ± 0.048    |    0.836 ± 0.029     |
> |    Avg. Node Coverage  |     0.398    |     0.533    |    0.668     |     0.749    |
> |    Avg. Runtime (sec)     |     60.268    |     83.238    |     139.939    |     211.601    |
>
> - **Why not provide experimental evidence to test RSNN generalization?** Thank you for this suggestion. We have conducted extensive empirical evaluations on molecular and protein benchmarks ranging from a few thousand to hundreds of thousands of graphs. Across these diverse datasets, RSNN consistently achieves higher test‐set performance than the RWNN baselines, providing strong practical confirmation of the tighter learning bounds established in our analysis.
>
> - **Why not compare to subgraph GNNs?** While we originally focused on RWNNs as our main baselines, you are right; we should also compare to subgraph GNNs. Thus, we have added the Equivariant Subgraph Aggregation Network (ESAN; Bevilacqua et al., 2022) to our experiments. ESAN generates one node‐deleted subgraph per node, applies a standard GNN to each, and aggregates the resulting embeddings via a DeepSets module to form the final graph representation. ESAN indeed outperforms RWNN under minimal coverage (m = 1), but RWNN matches or exceeds ESAN’s accuracy once full coverage is obtained (m = 16) (Table 2 in response to Reviewer k2UB). This empirical result aligns with our theoretical analysis that RWNNs with partial graph coverage are less expressive than subgraph GNNs. RSNN on the other hand, achieves full node coverage and high edge coverage with just a single search, outperforming both RWNN and ESAN even when m=1. We will incorporate ESAN comparisons into the main text to enrich our empirical evaluation.
>
> **References**
>
> Said, A., Bayrak, R., Derr, T., Shabbir, M., Moyer, D., Chang, C., & Koutsoukos, X. (2023). Neurograph: Benchmarks for graph machine learning in brain connectomics. Advances in Neural Information Processing Systems, 36, 6509-6531.
>
> Bevilacqua, B., Frasca, F., Lim, D., Srinivasan, B., Cai, C., Balamurugan, G., & Maron, H. (2022). Equivariant subgraph aggregation networks. International Conference on Learning Representations 2022

---

> > ### Comment · Reviewer_jLHR · 2025-08-01
> >
> > Thank you very much for your reply, it solved my problem very well. Based on this model's practicality, I have decided to maintain my positive rating.

---

> ### Author Response · Authors · 2025-08-05
>
> Thank you for your positive feedback! Your suggestions have significantly improved the quality of our work. We will be sure to incorporate each of them in the revised manuscript.

---

### Official Review · Reviewer_k2UB · 2025-07-01

**Clarity:** 3
**Significance:** 3
**Originality:** 3
**Rating:** 5
**Confidence:** 4

**Summary:**

This paper proposes RSNN, a graph representation learning method based on depth-first search (DFS). To address the limited expressivity of RWNNs under realistic sampling constraints—due to insufficient node and edge coverage—RSNN generates spanning trees via DFS to achieve full graph coverage with significantly lower sampling cost. Theoretical and empirical results demonstrate that RSNN offers stronger expressivity and better performance, especially on sparse graphs such as molecular and protein datasets.

**Questions:**

1. The experiments focus on sparse molecular and protein graphs. How does RSNN perform on larger or denser graphs? Additional results or discussion would clarify its generality.

2. Since subgraph GNNs are discussed theoretically, can the authors add empirical comparisons with such methods to better position RSNN?

3. The paper lacks detailed explanation of methods, such as the DFS sampling and spanning tree encoding. Please clarify or provide pseudo-code to improve reproducibility.

**Ethical Concerns:**

["NO or VERY MINOR ethics concerns only"]

**Final Justification:**

The authors have addressed my concerns, and I will maintain my positive score.

**Limitations:**

yes

**Quality:**

3

**Strengths And Weaknesses:**

Strengths:

1. Novel analytical perspective: The paper analyzes model expressivity from the perspective of node and edge coverage, providing a new angle for understanding the limitations of RWNNs.

2. Theoretical support: It offers solid theoretical results, showing how full coverage improves RWNN performance and establishing universal approximation and generalization guarantees for RSNNs.

3. The paper is clearly written and well-structured, making the motivation, methodology, and analysis easy to follow.

4. Experiments on molecular and protein benchmarks demonstrate consistent and meaningful improvements over RWNNs, using fewer sampled sequences.

Weaknesses:

1. The experimental evaluation focuses mainly on small and relatively sparse datasets (molecular and protein graphs), leaving it unclear how well the proposed method generalizes to larger or denser graphs.

2. Some technical details of the method are presented at a high level, and certain design choices (e.g., how DFS is implemented or integrated with the sequence model) are not clearly explained, making it harder to assess reproducibility and implementation complexity.

---

> ### Author Rebuttal · Authors · 2025-07-30
>
> Thank you very much for your thoughtful feedback! We are delighted that  you found our coverage perspective novel, our theoretical results compelling, our experiments consistent and meaningful, and our paper overall clearly written and well-structured. Below, we address your specific comments individually.
>
> - **How does RSNN perform on larger or denser graphs?** Great question. We ran additional experiments to measure how RSNN scales to **dense** graphs (see experiments below). We don’t evaluate on large graphs because we focus our evaluation on small to medium graphs, where global structure, long-range dependencies, and high expressivity are essential, in comparison to large graphs, where locality is more important to capture and as a result message-passing GNNs already work well (e.g., node classification in citation networks). To demonstrate that RSNN generalizes beyond sparse molecular and protein graphs to **denser** graphs, we have since evaluated it on two graph classification benchmarks of dense brain networks from NeuroGraph (Said et al., 2023): a Gender prediction set of 1000 graphs (avg. 1000 nodes, avg. 46000 edges, avg. degree ≈ 46) and an Activity prediction set of 7500 graphs (avg. 400 nodes, avg. 7000 edges, avg. degree ≈ 18), both substantially denser than our molecular and protein datasets. We compare performance and runtime of RSNN vs. CRAWL, the strongest RWNN baseline. RSNN consistently outperforms CRAWL on both tasks, for all $m$, confirming its ability to leverage global structure in dense regimes (Table 1), underscoring that RSNN retains its performance advantage on denser graphs.
>
> Table 1. Mean and standard deviation of CRAWL and RSNN performance on dense NeuroGraph datasets.
> | | Gender (ACC) | Activity (ACC) |
> | ------- | ------- | ------- |
> | (m=1) CRAWL | 0.640 ± 0.021 |    0.614 ± 0.007     |
> | (m=1) RSNN | **0.662 ± 0.027**    |     **0.619 ± 0.020**    |
> | (m=4) CRAWL | 0.676 ± 0.023 |    0.763 ± 0.009     |
> | (m=4) RSNN | **0.693 ± 0.008**    |    **0.797 ± 0.014**     |
> | (m=16) CRAWL | 0.676 ± 0.033    |     0.850 ± 0.004    |
> | (m=16) RSNN |    **0.707 ± 0.023**    |     **0.865 ± 0.007**    |
>
> - **Can authors add empirical comparisons with Subgraph GNNs?** To better position RSNN against subgraph GNNs empirically, we have included the Equivariant Subgraph Aggregation Network (ESAN; Bevilacqua et al., 2022) as a subgraph‐GNN baseline. ESAN operates by constructing node‐deleted subgraphs, removing each node and its incident edges, applying a standard GNN encoder to each subgraph, and then pooling the resulting embeddings with a DeepSets module to yield a global graph representation. In our experiments (Table 2), ESAN can indeed outperform RWNN when RWNN only achieves partial coverage sampling only one walk (m = 1). However, once RWNNs attain full coverage (m = 16), they match or exceed ESAN’s performance. This aligns with our theoretical analysis that partial‐coverage RWNNs are less expressive than subgraph GNNs. RSNN on the other hand, achieves full node coverage and high edge coverage with just a single search, outperforming both RWNN and ESAN, even when m=1. We will incorporate the ESAN baseline into the main paper to strengthen our empirical evaluation.
>
> Table 2. Mean and standard deviation of RWNN, RSNN, ESAN performance on SIDER and BBBP MoleculeNet datasets.
> | | SIDER (AUC) | BBBP (AUC) |
> | ------- | ------- | ------- |
> | ESAN | 0.660 ± 0.022 |    0.741 ± 0.046     |
> | (m=1) RWNN | 0.622 ± 0.029 |    0.736 ± 0.073     |
> | (m=1) RSNN | **0.665 ± 0.020**    |   **0.866 ± 0.023**     |
> | (m=16) RWNN | 0.664 ± 0.027    |     0.860 ± 0.017    |
> | (m=16) RSNN | **0.683 ± 0.027**    |     **0.886 ± 0.025**    |
>
> - **Implementation Details.** In the revision, we will expand Section 4 to include a comprehensive description of the DFS sampling procedure and its integration with the sequence model, based on the pseudocode from Appendix A.2. We will also present the Algorithms in the main text, showing both the random-walk and random-search sampling routines along with clear annotations on how they interface with the positional encodings (from Appendix A.1) and the sequence encoder. We will include a concise runtime and memory complexity comparison of RWNN, CRAWL, and RSNN. The full technical discussion, including extended implementation notes, will remain in Appendices A.1 and A.2 for readers seeking further details. Lastly, our code will also be made publicly available for full reproducibility.
>
> **References**
>
> Said, A., Bayrak, R., Derr, T., Shabbir, M., Moyer, D., Chang, C., & Koutsoukos, X. (2023). Neurograph: Benchmarks for graph machine learning in brain connectomics. Advances in Neural Information Processing Systems, 36, 6509-6531.
>
> Bevilacqua, B., Frasca, F., Lim, D., Srinivasan, B., Cai, C., Balamurugan, G., & Maron, H. (2022). Equivariant subgraph aggregation networks. International Conference on Learning Representations 2022

---

> > ### Comment · Reviewer_k2UB · 2025-08-07
> > **Rebuttal Response**
> >
> > I appreciate the author's efforts in the rebuttal, which have resolved my concerns. I will keep my initially assigned positive score.

---

> ### Comment · Area_Chair_AxjS · 2025-08-06
> **nudge from ac**
>
> Thanks for your review. Could you please respond to the author's rebuttal as soon as possible?
>
> Thanks
> AC

---

### Official Review · Reviewer_CoWK · 2025-07-01

**Clarity:** 3
**Significance:** 2
**Originality:** 2
**Rating:** 5
**Confidence:** 3

**Summary:**

This paper presents random search neural networks (RSNNs), which operate on random searches to provide a more efficient, expressive, and generalizable model compared to random walk neural networks (RWNNs). The authors provide a series of experimental results and theoretical justifications on the advantages of RSNNs.

**Questions:**

See weaknesses. Additionally:
- Tönshoff et al., 2023 state that GNNs intuitively have an advantage over RWNN methods such as CRaWl since GNNs effectively perform a breadth-first search. Given this and the fact that a final permutation-invariant function that is applied at the end, do the authors have some intuition as to if using BFS rather than DFS would meaningfully change the performance of RSNNs?

References:

Tönshoff et al. Walking out of the weisfeiler leman hierarchy: Graph learning beyond message passing. In _TMLR_, 2023.

**Ethical Concerns:**

["NO or VERY MINOR ethics concerns only"]

**Final Justification:**

The authors provide clear justification for their model design choices and will incorporate additional discussion around their theoretical results related to random walk encodings. They have also provided additional experimental results on dense graphs and compared to additional subgraph-GNN models which support their theoretical results.

**Limitations:**

Yes

**Paper Formatting Concerns:**

I have no formatting concerns.

**Quality:**

3

**Strengths And Weaknesses:**

Strengths:
- The paper is well written and easy to follow
- Experimental results support the claim that RSNNs improve performance across several different benchmarking tasks. Inclusion of traditional molecular fingerprinting methods provide a holistic comparison for molecular models.
- The authors provide clear theoretical analysis for RSNN - Theorem 4.2 in particular provides an interesting result.

Weaknesses:
- Although the authors present a compelling argument that RSNNs are more efficient in achieving full coverage of a graph, the following results linking higher coverage to increased expressivity seem somewhat trivial. Especially since most of the results are related to graph-level tasks, it is obvious that seeing more of the input graph would lead to better performance.
  - Theorem 3.1 also seems somewhat trivial given a recent result in Bao et al., 2025 (Theorem 4.6) which already proves that motif/homomorphism encodings (which are directly related to subgraphs) are more expressive than random walk encodings. Could the authors please address how their result meaningfully builds upon this?

References:

Bao et al. Homomorphism Counts as Structural Encodings for Graph Learning. In ICLR, 2025.

---

> ### Author Rebuttal · Authors · 2025-07-30
>
> Thank you for your detailed and valuable feedback. We’re encouraged that you found the paper well-written and easy to follow, our experimental results both holistic and compelling, and our theoretical analysis, particularly Theorem 4.2, insightful. Below, we address your specific comments individually.
>
> - **Coverage as an expressive bottleneck.** We agree that intuitively, “seeing more of the graph should help.” What our contribution adds is a *formal, quantitative* statement of how partial coverage limits expressivity even when the downstream sequence model is already universal and equipped with expressive positional encodings. By establishing the limitations induced by partial coverage, we give a rigorous justification for replacing random walks with our random-search sampler.
> - **How do theoretical results build on Bao et al., 2025 (Thm 4.6)?** Bao et al. compare *structural encodings*, showing that random-walk structural encoding (RWSE), defined as the diagonal entries of the $\ell$-step random walk matrix $(D^{-1}A)^\ell$, is strictly weaker than their motif-based encoding (MoSE) even when $\ell$ is infinite. In contrast, our Theorem 3.1 tackles a different axis: the *expressive power of full architectures*. We study RWNNs, whose inputs are the full length-$\ell$ walk **sequences**, not just the return probabilities along the diagonals of the random walk matrix. Consequently, an RWNN equipped with a universal sequence model is fundamentally more expressive than the RWSE structural encoding. Theorem 3.1 shows that this stronger class can become strictly less expressive than subgraph-based GNNs whenever the walk sampler fails to achieve full coverage. In other words, we pinpoint coverage of the sampling scheme rather than the capacity of the downstream sequence model as the limiting factor. This complements Bao et al., 2025: their result isolates a limitation of a *particular structural encoding*, whereas ours links expressivity to the *search strategy itself* and motivates richer samplers such as our random-search procedure. We will add a discussion after Theorem 3.1 to make this connection explicit and clarify that our contribution extends the RWSE vs MoSE comparison to the architectural level and to the role of graph coverage.
> - **BFS vs. DFS.** We adopt a depth-first search (DFS) sampler rather than breadth-first search (BFS) because DFS preserves continuity in sequences which is important in reflecting the graph structure. In contrast, BFS jumps within neighborhoods and yields many disconnected subsequences. Empirically, we observe that swapping DFS for BFS results in a slight decrease in performance. We will add this justification for utilizing DFS rather than BFS when introducing our RSNN approach.

---

> > ### Comment · Reviewer_CoWK · 2025-08-04
> >
> > I thank the authors for their response - they have addressed most of my concerns. For completeness sake, could they please provide the DFS vs BFS results that they obtained? It would be interesting to see those results and include them in the Appendix.

---

> > > ### Author Response · Authors · 2025-08-04
> > >
> > > Certainly. Below, we include our results on RSNN with DFS versus BFS sampling on the molecular datasets and observe that DFS yields slightly higher performance on five of them, likely because it better preserves connectivity and continuity in the generated sequences. We will include these detailed results in the Appendix for completeness.
> > >
> > > Table 3. Mean and standard deviation of RSNN (DFS) and RSNN (BFS) on molecular datasets.
> > > | | CLINTOX | SIDER | BACE | TOXCAST | BBBP | TOX21 |
> > > | ------- | ------- | ------- | ------- | ------- | ------- | ------- |
> > > | RSNN (BFS) | 0.849 ± 0.062 | 0.660 ± 0.029 | 0.781 ± 0.023 | 0.746 ± 0.029 | **0.879 ± 0.023** | 0.790 ± 0.018 |
> > > | RSNN (DFS) | **0.874 ± 0.031** | **0.665 ± 0.020** | **0.800 ± 0.024** | **0.763 ± 0.026** | 0.866 ± 0.023 | **0.801 ± 0.017** |

---

> > > > ### Comment · Reviewer_CoWK · 2025-08-04
> > > >
> > > > Thank you for your reply - given the additional results presented here (and for dense graphs and subgraph-based GNNs as requested by the other reviewers), I am happy to see this paper accepted and have increased my score accordingly.

---

> ### Author Response · Authors · 2025-08-05
>
> Thank you for your positive and constructive feedback and for increasing your score! Your suggestions have greatly improved our work, and we will incorporate all of them in the revised manuscript.

---

### Official Review · Reviewer_kbHS · 2025-07-22

**Clarity:** 3
**Significance:** 4
**Originality:** 4
**Rating:** 5
**Confidence:** 4

**Summary:**

The paper compares several methods for obtaining data about graphs via neural networks: in one approach (subgraphs GNN), the neural network is based on subgraphs, in the other (RWNN), it is based on random walks, and in the third (RSNN), it is based on random depth-first trees. The paper presents several theorems, as well as empirical results, suggesting that RSNN is superior to RWNN, which in turn is superior to subgraphs GNN. I suggest accepting this paper.

**Questions:**

.

**Ethical Concerns:**

["NO or VERY MINOR ethics concerns only"]

**Final Justification:**

The authors answered my concerns to my satisfaction

**Limitations:**

.

**Paper Formatting Concerns:**

There are no paper formatting concerns.

**Quality:**

3

**Strengths And Weaknesses:**

The results are strong, but it is very hard for the reader to understand what they really are, because too many notions are not well-defined. For example, the notion of expressivity is presented many times, but never rigorously defined. In line 146, there is a list of papers evaluating expressivity, but no hint as to what this expressivity is or how it is evaluated. In Theorem 3.1, it is unclear what "equipped with maximally expressive components" means, and so on. It would help a lot to give a definition, or at least a heuristic explanation.

From Proposition 2.1, I guess expressivity has to do with distinguishing between nonisomorphic graphs, and that "appropriate graph metric" means a metric that somehow measures how graphs are close to being isomorphic. But I would be very happy with a few words affirming or refuting my guess.

---

> ### Author Rebuttal · Authors · 2025-07-30
>
> We appreciate your detailed and thoughtful feedback and comments. Below, we address your specific comments individually.
> - **Does expressivity have to do with distinguishing between nonisomorphic graphs?** Yes; proposition 2.1 adopts the classical Weisfeiler–Lehman perspective, defining an RWNN as expressive when it distinguishes pairs of non-isomorphic graphs. Theorems 3.1 and 3.2 use the functional perspective, where a model is expressive if it can approximate the space of continuous graph-level functions $f : \mathcal{G} \to \mathbb{R}^{d}$. These perspectives coincide for GNNs: any architecture that separates isomorphism classes is also a universal approximator on continuous graph functions, and vice versa (Chen et al., 2019). We will emphasize this equivalence in the updated version and state explicitly that Proposition 2.1 relies on the isomorphism-distinguishing criterion, whereas Theorems 3.1 and 3.2 rely on the universal-approximation criterion.
> - **In Theorem 3.1, what does "equipped with maximally expressive components" mean?** By “maximally expressive components,” we mean that given sufficient width/depth, every learnable sub-module can approximate any continuous function (i.e., is universal) defined on its domain. Theorem 3.1 assumes each such component has enough capacity to reach this universal regime; thus any remaining approximation error is due to the sampling scheme. We will make this definition explicit in the revised manuscript.
> - **Does "appropriate graph metric" mean a metric that somehow measures how graphs are close to being isomorphic?** Yes; In the proofs, we instantiate the metric with the graph-edit distance, which measures how close graphs are to being isomorphic (Keriven and Pyre 2019). We will explicitly state our use of the graph edit distance in the revised statement of Theorem 4.2.
>
> **References**
>
> Chen, Z., Villar, S., Chen, L., & Bruna, J. (2019). On the equivalence between graph isomorphism testing and function approximation with gnns. Advances in neural information processing systems, 32.
>
> Keriven, N., & Peyré, G. (2019). Universal invariant and equivariant graph neural networks. Advances in neural information processing systems, 32.

---

> > ### Comment · Reviewer_kbHS · 2025-08-09
> > **dear authors**
> >
> > Your comments made things clearer. I changed the scores for clarity and the rating accordingly.

---

> ### Comment · Area_Chair_AxjS · 2025-08-06
> **nudge from ac**
>
> Thanks for your review. Could you please respond to the author's rebuttal as soon as possible?
>
> Thanks
> AC

---

> > ### Author Response · Authors · 2025-08-08
> >
> > Thank you for all your great feedback. Since we're less than 24 hours from the end of the discussion period, we just wanted to check in and see if you had any more questions. We are very happy to address any remaining questions you might have about the paper.

---

### Decision · Program_Chairs · 2025-09-17

**Decision:**

Accept (poster)

**Comment:**

This paper suggest a new graph neural network based on graph random walks. The method's novelty is in its ability to achieve full node coverage faster than existing methods. Analysis of the expressive power of the model is provided, as well as strong empirical results. The reviewers were positive about the results in this paper. Initially there were some concerns regarding mathematical terminology and related works, but these concerns were addressed during the rebuttal. I recommend acceptance.